# WHAT TO EXPECT OF HARDWARE METRIC PREDICTORS IN NEURAL ARCHITECTURE SEARCH

## ABSTRACT

Modern Neural Architecture Search (NAS) focuses on finding the best performing architectures in hardware-aware settings; e.g., those with an optimal tradeoff of accuracy and latency. Due to many advantages of prediction models over live measurements, the search process is often guided by estimates of how well each considered network architecture performs on the desired metrics. Typical prediction models range from operation-wise lookup tables over gradient-boosted trees and neural networks, with little known information on how they compare. We evaluate 18 different performance predictors on ten combinations of metrics, devices, network types, and training tasks, and find that MLP models are the most promising. We then simulate and evaluate how the guidance of such prediction models affects the subsequent architecture selection. Due to inaccurate predictions, the selected architectures are generally suboptimal, which we quantify as an expected reduction in accuracy and hypervolume. We show that simply verifying the predictions of just the selected architectures can lead to substantially improved results. Under a time budget, we find it preferable to use a fast and inaccurate prediction model over accurate but slow live measurements.

## 1 INTRODUCTION

Modern neural network architectures are designed not only considering their primary objective, such as accuracy. While existing architectures can be scaled down to work with the limited available memory and computational power of, e.g., mobile phones, they are significantly outperformed by specifically designed architectures (Howard et al., 2017; Sandler et al., 2018; Zhang et al., 2018; Ma et al., 2018). Standard hardware metrics include memory usage, number of model parameters, Multiply-Accumulate operations, energy consumption, latency, and more; each of which may be limited by the hardware platform or network task. As the range of tasks and target platforms grows, specialized architectures and the methods to find them efficiently are gaining importance.

The automated design and discovery of specialized architectures is the main intent of Neural Architecture Search (NAS). This recent field of study repeatedly broke state of the art records (Zoph et al., 2018; Real et al., 2018; Cai et al., 2019; Tan & Le, 2019; Chu et al., 2019a; Hu et al., 2020) while aiming to reduce the researchers' involvement with this tedious and time-consuming process to a minimum. As the performance of each considered architecture needs to be evaluated, the hardware metrics need to be either measured live or guessed by a trained prediction model. While measuring live has the advantage of not suffering from inaccurate predictions, the corresponding hardware needs to be available during the search process. Measuring on-demand may also significantly slow down the search process and necessitates further measurements for each new architecture search. On the other hand, a prediction model abstracts the hardware from the search code and simplifies changes to the optimization targets, such as metrics or devices. The data set to train the predictor also has to be collected only once so that a trained predictor then works in the absence of the hardware it is predicting for, e.g., in a cloud environment. Furthermore, a differentiable predictor can be used for gradient-based architecture optimization of typically non-differentiable metrics (Cai et al., 2019; Xu et al., 2020; Nayman et al., 2021).

While the many advantages make predictors a popular choice of hardware-aware NAS (e.g. Xu et al. (2020); Wu et al. (2019); Wan et al. (2020); Dai et al. (2020); Nayman et al. (2021)), there are no guidelines on which predictors perform best, how many training samples are required, or

what happens when a predictor is inaccurate. This work investigates the above points. As a first contribution, we conduct large-scale experiments on ten hardware-metric datasets chosen from HW-NAS-Bench (Li et al., 2021a) and TransNAS-Bench-101 (Duan et al., 2021). We explore how powerful the different predictors are when using different amounts of training data and whether these results generalize across different network architecture types. As a second contribution, we extensively simulate the subsequent architecture selection to investigate the impact of inaccurate predictors. Our results demonstrate the effectiveness of network-based prediction models; provide insights into predictor mistakes and what to expect from them. To facilitate reproducibility and further research, our experimental results and code are made available in Appendix A.

## 2 RELATED WORK

**NAS Benchmarks:** As the search spaces of NAS methods often differ from one another and lack extensive studies, the difficulty of fair comparisons and reproducibility have become a major concern (Yang et al., 2019; Li & Talwalkar, 2020). To alleviate this problem, researchers have exhaustively evaluated search spaces of several thousand architectures to create benchmarks (Ying et al., 2019; Dong & Yang, 2020; Dong et al., 2020; Siems et al., 2020), containing detailed statistics for each architecture. TransNAS-Bench-101 (Duan et al., 2021) evaluates several thousand architectures across seven diverse tasks and finds that the best task-specific architectures may vary significantly.

The popular NAS-BENCH 201 benchmark (Dong & Yang, 2020) has been further extended with ten different hardware metrics for all 15625 architectures on each of the three data sets CIFAR10, CIFAR100 (Krizhevsky et al., 2009) and ImageNet16-120 (Chrabaszcz et al., 2017). Major findings of this HW-NAS Bench (Li et al., 2021a) include that FLOPs and the number of parameters are a poor approximation for other metrics such as latency. Many existing NAS methods use such inadequate substitutes for their simplicity and would benefit from their replacement with better prediction models. Li et al. also find that hardware-specific costs do not correlate well across hardware platforms. While accounting for each device's characteristics improves the NAS results, it is also expensive. Predictors can reduce costs by requiring fewer measurements and shorter query times. [1].

**Predictors in NAS:** Aside from real-time measurements (Tan et al., 2019; Yang et al., 2018), hardware metric estimation in NAS is commonly performed via Lookup Table (Wu et al., 2019), Analytical Estimation or a Prediction Model (Dai et al., 2020; Xu et al., 2020). While an operation- and layer-wise Lookup Table can accurately estimate hardware-agnostic metrics, such as FLOPs or the number of parameters (Cai et al., 2019; Guo et al., 2020; Chu et al., 2019a), they may be suboptimal for device-dependent metrics. Latency and energy consumption have non-obvious factors that depend on hardware specifics such as memory, cache usage, the ability to parallelize each operation, and an interplay between different network operations. Such details can be captured with neural networks (Dai et al., 2020; Mendoza & Wang, 2020; Ponomarev et al., 2020; Xu et al., 2020) or other specialized models (Yao et al., 2018; Wess et al., 2021).

Of particular interest is the correct prediction of the model loss or accuracy, possibly reducing the architecture search time by orders of magnitude (Mellor et al., 2020; Wang et al., 2021; Li et al., 2021b). In addition to common predictors such as Linear Regression, Random Forests (Liaw et al., 2002) or Gaussian Processes (Rasmussen, 2003); specialized techniques may exploit training curve extrapolation, network weight sharing or gradient information. Our experiments follow the recent large-scale study of White et al. (2021), who compare 31 diverse accuracy prediction methods based on initialization and query time, using three NAS benchmarks.

## 3 PREDICTING HARDWARE METRICS

Our methods follow the large-scale study of White et al. (2021), who compared a total of 31 accuracy prediction methods. The differences between accuracy and hardware-metric prediction, our selection of predictors, and the general training pipeline are described in this section. In our experiments on HW-NAS-Bench and TransNAS-Bench-101, described in Section 4, we then compare these predictors across different training set sizes.

---

[1]For further reading, we recommend a recent survey on hardware-aware NAS (Benmeziane et al., 2021)

**Differences to accuracy predictors:** There are fundamental differences when predicting hardware metrics and the accuracy of network topologies. The most essential is the cost to obtain a helpful predictor, which may vary widely for accuracy prediction methods. While determining the test accuracy requires the costly and lengthy training of networks, measuring hardware metrics does not necessitate any network training. Consequentially, specialized accuracy-estimation methods that rely on trained networks, loss history, learning curve extrapolation, or early stopping do not apply to hardware metrics. Furthermore, so-called zero-cost proxies that predict metrics from the gradients of a single batch are dependant on the network topology but not on the hardware the network is placed on. Therefore, the dominant hardware-metric predictor family is model-based.

Since all relevant predictors are model-based, they can be compared by their training set size. This simplifies the initialization time of a predictor as the number of prior measured architectures on which they are trained. In stark contrast, some accuracy predictors do not need any training data, while others require several partially or fully trained networks. Since an untrained network and a few batches suffice to measure a hardware-metric, the collection of such a training set is comparably inexpensive.

Additionally, hardware predictors are generally used supplementary to a one-shot network optimized for loss or accuracy. Depending on the NAS method, a fully differentiable predictor is required in order to guide the gradient-based architecture selection. Typical choices are Lookup Tables (Cai et al., 2019; Nayman et al., 2021) and neural networks (Xu et al., 2020).

**Model-based predictors:** The goal of a predictor $f_p(a)$ is to accurately approximate the function $f(a)$, which may be, e.g., the latency of an architecture $a$ from the search space $\mathcal{A}$. A model-based predictor is trained via supervised learning on a set $\mathcal{D}_{train}$ of datapoints $(a, f(a))$, after which it can be inexpensively queried for estimates on further architectures. The collection of the dataset and the duration of the training are referred to as *initialization time* and *training time* respectively.

The quality of such a trained predictor is generally determined by the (ranking) correlation between measurements $\{f(a)|a \in \mathcal{A}_{test}\}$ and predictions $\{f_p(a)|a \in \mathcal{A}_{test}\}$ on the unseen architectures $\mathcal{A}_{test} \subset \mathcal{A}$. Common correlation metric choices are Pearson (PCC), Spearman (SCC) and Kendall's Tau (KT) (Chu et al., 2019b; Yu et al., 2020; Siems et al., 2020).

Our experiments include 18 model-based predictors from different families: Linear Regression, Ridge Regression (Saunders et al., 1998), Bayesian Linear Regression (Bishop, 2007), Support Vector Machines (Cortes & Vapnik, 1995), Gaussian Process (Rasmussen, 2003), Sparse Gaussian Process (Candela & Rasmussen, 2005), Random Forests (Liaw et al., 2002), XGBoost (Chen & Guestrin, 2016), NGBoost (Duan et al., 2020), LGBoost (Ke et al., 2017), BOHAMIANN (Springenberg et al., 2016), BANANAS (White et al., 2019), BONAS (Shi et al., 2020), GCN (Wen et al., 2020), small and large Multi-Layer-Perceptrons (MLP), NAO (Luo et al., 2018), and a layer-operation-wise Lookup Table model. We provide further descriptions and implementation details in Appendix B.

**Hyper-parameter tuning:** The default hyperparameters of the used predictors vary significantly in their levels of hyper-parameter tuning, especially in the context of NAS. Additionally, some predictors may internally make use of cross-validation, while others do not. Following White et al. (2021), we attempt to level the playing field by running a cross-validation random-search over hyper-parameters each time a predictor is fit to data. Each search is limited to 5000 iterations and a total run time of 15 minutes and naturally excludes any test data. The predictor-specific parameter details are given in Appendix C.

**Training pipeline** To make a reliable comparison, we use the NASLib library (Ruchte et al. (2020), see Appendix A). We fit each predictor on each dataset and training size 50 times, using seeds $\{0, ..., 49\}$.

Some predictors internally normalize the training values (subtract mean, divide by standard deviation). We choose to explicitly do this for all predictors and datasets, which reduces the dependency of hyper-parameters (e.g. learning rate) on the dataset and allows us to analyze and compare the prediction errors across datasets more effectively.

## 4 PREDICTOR EXPERIMENTS

We compare the different predictor models based on two NAS benchmarks, HW-NAS-Bench (Li et al., 2021a) and TransNAS-Bench-101 (Duan et al., 2021). They differ considerably by their network tasks, hardware devices, and architecture designs.

**HW-NAS-Bench architecture design and datasets**  In HW-NAS-Bench, each architecture is solely defined by the topology of a building block ("cell"), which is stacked multiple times to create a complete network. Each cell is completely defined by choosing six candidate operations. Since they select from five different candidates each time, there are $5^6 = 15625$ unique cell topologies. These cells are not fully sequential but contain paths of different lengths, which is visualized in Appendix D.

HW-NAS-Bench provides ten hardware statistics on CIFAR10, CIFAR100 Krizhevsky et al. (2009) and ImageNet16-120 Chrabaszcz et al. (2017), of which we exclude the incomplete EdgeTPU metric. Thus there are 27 data sets of varying difficulty. As detailed in Appendix E, 12 of them can be accurately fit with Linear Regression and only 25 training samples. Many are also very similar since their measured networks differ only by the number of image classes. We therefore select five datasets that (1.) are not trivial to learn as they are non-linear and (2.) not redundant:

- ImageNet16-120, raspi4, latency
- CIFAR100, pixel3, latency
- CIFAR10, edgegpu, latency
- CIFAR100, edgegpu, energy consumption
- ImageNet16-120, eyeriss, arithmetic intensity

**TransNAS-Bench-101 architecture design and datasets**  TransNAS-Bench-101 contains information for 7,352 different network architectures, used as backbones in seven diverse vision tasks. Since 4,096 are also a subset of HW-NAS-Bench, we focus on the remaining 3,256 architectures with a macro-level search space. Unlike a micro-level search space, where a cell is stacked multiple times to create a network, each network layer and block is considered individually. In particular, the TransNAS-Bench-101 networks consist of four to six pairs of ResNet blocks (He et al., 2016), which may modify the image size and channels in four ways: not at all, double the channel count, halve the spatial size, and both. Every network has to double the channel count 1 to 3 times, resulting in 3,256 unique architectures. The networks may consequentially differ in their number of layers (depth), the number of channels (width), and image size at any layer.

As done for HW-NAS-Bench, we select five of the seven available datasets for their latency measurements. Aside from the self-supervised Jigsaw task, there is little difference between the cross-task latency measurements (see Appendix E). We evaluate the possibly redundant datasets nonetheless, since latency predictions in macro-level search spaces are an important domain for NAS on image classification and object detection tasks:

- Object classification
- Scene classification
- Room layout
- Jigsaw
- Semantic segmentation

**Fitting results and comparison**  The results, averaged over all selected HW-NAS-Bench and TransNAS-Bench-101 datasets, are presented in Figures 1a and 1b, respectively. The left plots present the absolute predictor performance, the right ones make relative comparisons easier.

Unsurprisingly, more training samples (i.e., evaluated architectures) generally lead to better prediction results, even until the entire search space is known (aside from the test set). This is true for most of the predictors, although e.g. Gaussian Processes and BOHAMIANN saturate early. The simple Linear Regression and Ridge Regression models also fail to make proper use of hundreds of data points but perform decently when only a few training samples are available. Interestingly, the same is true for the graph-encoding network-based predictors BONAS and GCN. While knowing how the different paths within each cell connect (see Appendix B) is especially useful given less than fifty training samples, the advantage disappears afterward. In contrast, the graph-encoding encoder-decoder approach of NAO performs decently at all times.

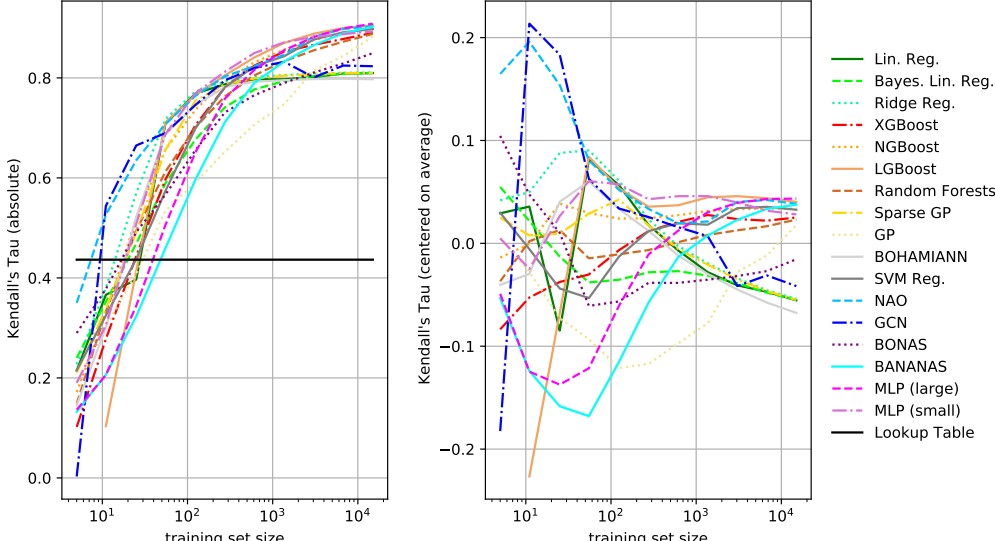

(a) Results on HW-NAS-Bench. NAO performs decently at all times, and none of the prediction models requires more than 60 training samples to improve over a Lookup Table model.

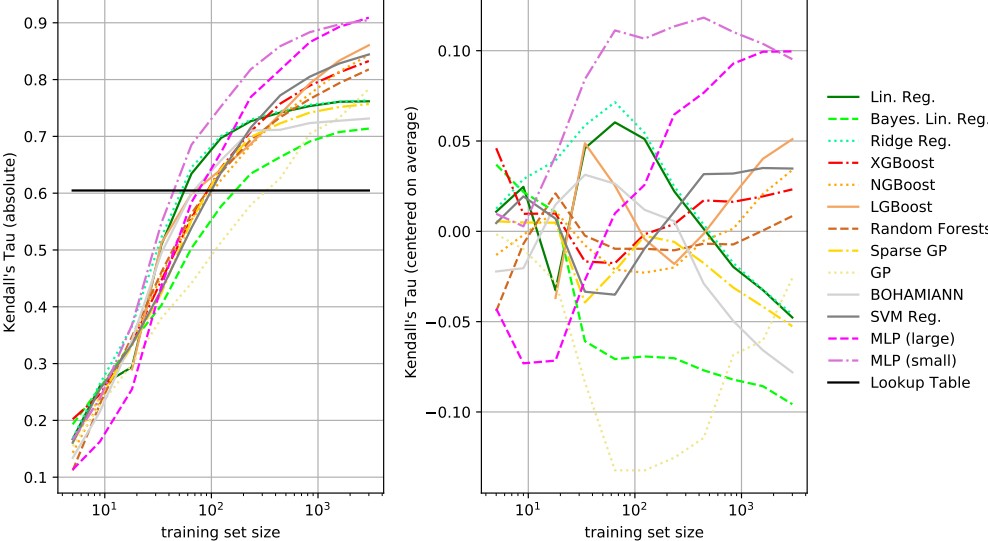

(b) Results on TransNAS-Bench-101. Since all network architectures are purely sequential by design, we do not evaluate predictors that specifically encode the architecture connectivity (BANANAS, BONAS, GCN, NAO). After as few as 20 training samples, MLP models outclass all other predictors.

Figure 1: How well the different predictors rank the test architectures, depending on the training set size and averaged over the five selected datasets. **Left plots:** absolute Kendall's Tau ranking correlation, higher is better. **Right plots:** same as left, but centered on the predictor-average.

Due to their powerful rule-based approach, tree-based models perform much better given many training samples. Under such circumstances, LGBoost is a candidate for the best predictor model. Similarly, the predictions of Support Vector Machines also benefit strongly from more samples.

The model we find to perform best for most training set sizes are MLPs. They are among the top predictors at almost all times in the HW-NAS-Bench, although tree-based models are competitive given enough data. After around 3,000 training samples, thinner and deeper MLPs improve over the wider and smaller ones. The path-encoding BANANAS model behaves similarly to a regular large MLP but requires more samples to reach the same performance. This is interesting since, aside from the data encoding, BANANAS is an ensemble of three large MLP models. Even though only the first network layer is affected by the data encoding, the more complicated path-encoding proves harmful

| | HW-NAS-Bench | | | | | TransNAS-Bench-101 |
|---|---|---|---|---|---|---|
| | Raspi4 | FPGA | Eyeriss | Pixel3 | EdgeGPU | Tesla V100 |
| latency | 0.45 (0.75) | 0.99 (0.97) | 0.99 (0.96) | 0.49 (0.78) | 0.21 (0.79) | 0.60 (0.70) |
| energy | | 0.99 (0.97) | 1.00 (0.99) | | 0.23 (0.79) | |
| arithmetic_intensity | | | 0.84 (0.81) | | | |

Table 1: The Kendall's Tau correlation of Lookup Tables and Linear Regression (in brackets, using only 124 training samples) across metrics and devices. Lookup Tables perform only marginally better on the FPGA and Eyeriss devices, but considerably worse in all other cases. More detailed statistics are available in Appendix E.

when the connectivity of the architectures in the search space is fixed. On TransNAS-Bench-101, MLP perform exceptionally well. They are much better than any other tested predictor once more than just 20 training samples are available. The small MLP model can achieve a KT correlation of 80% with just 200 training samples, which takes the best non-network-based predictor (Support Vector Machine) four times as many. They are also the only models that achieve a KT correlation of over 90%, about 5% higher than the next best model (LGBoost).

Finally, the Lookup Table models (black horizontal lines) perform poorly in comparison to any other predictor. Even though building such a model for HW-NAS-Bench datasets requires only 25 neighbored architectures, NAO and GCN perform better after just ten random samples. More than half of the predictor models require less than 25 random samples, while the worst need at most 60. On TransNAS-Bench-101, Lookup Tables perform comparably better. Building one requires only 21 neighbored architectures, and it takes most models between 50 and 100 random training samples to achieve better performance. When measured on a per dataset basis, we find that the Lookup Table models display a severe performance difference ranging from about 20% KT correlation (cifar10-edgegpu_latency and Jigsaw) to over 70% (ImageNet16-120-eyeriss_arithmetic_intensity and Semantic Segmentation, see Appendix E). Other models prove to be much more stable.

**Devices and Metrics**  The previously described results are based on a specific selection of HW-NAS-Bench and TransNAS-Bench-101 datasets that are hard to fit for Lookup Table models. As shown in Table 1, that is not always the case. The FPGA and Eyeriss hardware devices are very suitable for Lookup Tables, achieving an almost perfect ranking correlation is possible. Nonetheless, Linear Regression requires only 124 training samples to compete even there and is significantly better in every other case. We finally observe that the difficulty of fitting predictors primarily depends on the hardware device, much more than the measured metric.

## 5 EVALUATING THE PREDICTOR-GUIDED ARCHITECTURE SELECTION

Although the experiments in Section 4 greatly assist us in selecting a predictor, it is not clear what a specific Kendall's Tau correlation implies for the subsequent architecture selection. Given a perfect hardware metric predictor (Kendall's Tau = 1.0), we can expect that an ideal architecture search process will select the architectures with the best tradeoff of accuracy and the hardware metric, i.e., the true Pareto front. On the other hand, imperfect predictions result in the selection of supposedly-best architectures that are wrongly believed to be better.

To study how hardware predictors affect NAS results, we extensively evaluate the selection of such supposedly-best architectures in simulation. This approach can evaluate any combination of predictor quality, test set size, and dataset, without the technical difficulties of obtaining actual predictor models that precisely match such requirements. Since the hardware and accuracy prediction models are usually independent and can be studied in isolation, we use ground-truth accuracy values in all cases.

**Simulating predictors**  The main challenge of the simulation is to quickly and accurately model predictor outputs. We base our simulation on how predictor-generated values deviate from their ground-truth targets on the test set, which is explained in Figure 2 and further detailed in Appendix G. Since the simulated deviations are similar to those of actual predictors, simulated predictions are obtained by drawing random values from this deviation distribution and adding them to the ground-truth hardware measurements.

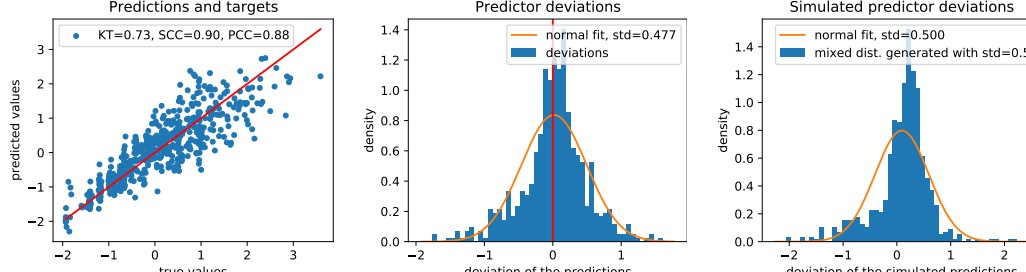

Figure 2: A trained XGBoost prediction model on normalized ImageNet16-120 raspi4-latency test data. **Left**: The latency prediction (y-axis) for every architecture (blue dot) is approximately correct (red line). **Center**: The same data as on the left, the distribution of deviations made by the predictor (blue) and a normal distribution fit to them (orange). **Right**: A mixed distribution can simulate typical deviation distributions as that in the center plot.

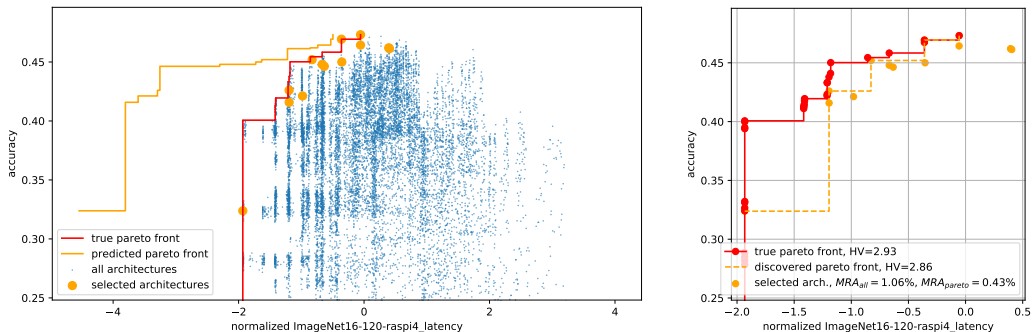

Figure 3: An example of predictor-guided architecture selection, std$=0.5$. **Left**: The simulated predictor makes an inaccurate latency prediction for each architecture (blue), resulting in the selection of the supposedly-best architectures (orange dots). Even though the predicted Pareto front (orange line) may differ significantly from the ground-truth Pareto front (red line), most selected architectures are close to optimal. **Right**: Same data as left. The true Pareto front (red) and that of the selected architectures (orange). Simply accepting all selected architectures results in a Mean Reduction of Accuracy (*MRA*) of $1.06\%$, while verifying the predictions and discarding inferior results improves that to $0.43\%$. The hypervolume (HV, area under the Pareto-fronts) is reduced by $0.07$.

A single example of a simulation can be seen in Figure 3. Although most selected architectures (orange) are close to the true optimum (red Pareto front), there almost always exists an architecture that has superior accuracy and, at most, the same latency. Simply accepting the 13 selected architectures in this particular example results in a mean reduction of accuracy ($MRA_{all}$) of $1.06\%$. In other words, the average selected architecture has $1.06\%$ lower accuracy than a comparable one on the true Paret front. However, simply verifying the hardware metric predictions through actual measurements reveals that some selected architectures are suboptimal. By choosing only the Pareto subset of the selection, the opportunity loss can be reduced to $0.43\%$ ($MRA_{pareto}$).

An important property of this approach is that it is independent of any particular optimization method. The supposedly-best architectures are always correctly identified, which is an upper bound on how well Bayesian Optimization, Evolutionary Algorithms, and other approaches can perform. The exemplary $MRA_{pareto}$ opportunity loss of $0.43\%$ is therefore unavoidable and depends solely on the hardware metric predictor, the dataset, and the number of considered architectures.

**Results** We simulate 1,000 architecture selections for each of the five chosen HW-NAS-Bench datasets, six different test set sizes, and eleven distribution standard deviations between $0.0$ and $1.0$. As exemplarily shown in Figure 3, each such simulation allows us to compute the mean reduction in accuracy (MRA) and the hypervolume (HV) under the Pareto fronts. The most important insights are visualized in Figure 4 and summarized below.

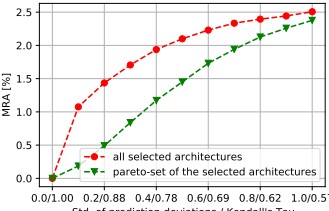 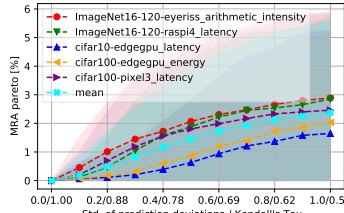 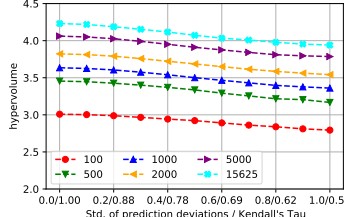

Figure 4: Simulation results, with the standard deviation of the predictor deviations and the resulting KT correlation on the x-axis. **Left**: Verifying the hardware predictions can significantly improve the results, even more so for better predictors. **Center**: The drops in average accuracy are dependant on the dataset and hardware metric. **Right**: Considering more candidate architectures and using better prediction models improves the results; larger values are better.

Verifying the predicted results matters (Figure 4, left). The best prediction models achieve a KT correlation of almost 0.9, which translates to a mean reduction in accuracy of $\text{MRA}_{all} \approx 1.5\%$. That means, for each selected architecture, there exists an architecture of equal or lower latency in the true Pareto set (if latency is the hardware metric) that improves the average accuracy by 1.5%. Even though all selected architectures are believed to form a Pareto set, that is not the case. Their optimal subset has a reduction of only $\text{MRA}_{pareto} \approx 0.5\%$, a significant improvement. However, finding this optimal subset requires actually measuring the hardware metrics of the architectures selected by the used NAS method.

Furthermore, the left of Figure 4 aids in anticipating the MRA given a specific predictor. If one used e.g. BOHAMIANN (KT≈0.8, see Figure1a) instead of MLPs or LGBoost (KT≈0.9), $\text{MRA}_{pareto}$ increases from around 0.5% to roughly 1.2%. The average accuracy of the selected architectures is thus reduced by another 0.7%, just by using an unsuitable hardware metric predictor. Lookup Tables (KT≈0.45) are not even visualized anymore, they have an $\text{MRA}_{pareto}$ of over 2.5%.

Another interesting observation is that the gap between $\text{MRA}_{all}$ and $\text{MRA}_{pareto}$ is wider for better predictors. This is a shortcoming of the MRA metric that we elaborate on in Appendix H.

The dataset and metric matter (Figure 4, center). While we generally present the results averaged over datasets, there exists some discrepancy among them. Most interestingly, predicting hardware metrics on harder classification problems (ImageNet16-120 is harder than CIFAR10) also results in a higher MRA. This is especially important since MRA is an absolute accuracy reduction. Even though the CIFAR10 networks achieve twice the accuracy of ImageNet16-120 networks, they lose less absolute accuracy to imperfect predictions. The order of MRA/dataset is primarily stable for any predictor KT correlation. Finally, as visualized by the shaded areas, the standard deviation of the MRA is generally huge. Consequentially, predictor-guided NAS is very likely to produce results of varying quality for each different predictor or search attempt, especially with less accurate predictors.

The number of considered architectures matters (Figure 4, right). We measure the hypervolume of the discovered Pareto front (i.e., the area beneath it, see Appendix H), which, unlike MRA, also considers the hardware metric. Quite obviously, if the architectures from the true Pareto set are not considered, they can not be selected. To achieve the highest possible hypervolume of around 4.2 (i.e. find the true Pareto set), every architecture in the search space must be evaluated with a perfect predictor. This is impossible in most real-world cases, where only a tiny fraction of all possible architectures can ever be considered.

For HW-NAS-Bench, considering 5000 architectures with perfect live measurements and predicting the metrics for all 15625 with ranking correlation KT≈0.73 results in selecting equivalent sets of architectures. As seen in Figure1a, Ridge Regression can achieve this performance with fewer than 100 training samples. Thus, a worse predictor leads to better results if it enables considering more architectures. This insight is especially crucial for live measurements, which are accurate but slow. Similarly, estimating the network accuracy with super-networks takes much more time than predicting their performance with a neural predictor (Wen et al., 2020). If the measurement of any metrics is the limiting factor, a guided selection of a cheap predictor is likely to do better.

## 6 DISCUSSION

**Chosen prediction methods**   Given the nature of hardware-metric prediction, only the subset of model-based predictors evaluated by White et al. (2021) is suitable. We extended this subset with four models, including the popular Lookup Table. We abstained from evaluating layer-wise predictors (e.g. Wess et al. (2021)) since such data is not available, and meta-learning predictors (Lee et al., 2021) due to the vast possibilities to configure them. A separate and specialized comparison between classic and meta-learning predictors seems favorable to us.

**Simulation limitations**   In contrast to evaluating real predictors, the simulation allows us to quickly make statements for any test set sizes and predictor-inaccuracies. However, naturally, the results are only approximations. While they match actual values, they are generally slightly pessimistic (see Appendix I). We also limit the simulation to HW-NAS-Bench since the changes to classification results are more accessible to interpretation than changes to loss values across different problem types. Finally, the current simulation approach can not investigate methods that absolutely require a trained one-shot network, such as gradient-based approaches. Including such methods is an interesting direction for future research.

**Transferability of the results**   Our evaluation includes five challenging and diverse datasets based on the micro-level search space of HW-NAS-Bench and five latency-based datasets of various macro-level search space architectures in TransNAS-Bench-101. Nonetheless, we find shared trends: All tested prediction models improve over Lookup Tables with little amounts of training data. Furthermore, most predictors benefit from more training data, even until the entire search space (aside from the test set) is known. We also find that network-based predictors are generally best but may be challenged by tree-based predictors if enough training data is available. Given only a few samples, Ridge Regression performs better than most other models.

**Recommendations**   While Lookup Tables are a cheap, simple, and popular model in gradient-based architecture selection, we find a significant variance in performance across tasks and devices (see Table 1 and Appendix E). We recommend replacing such models with either MLPs or Ridge Regression, which are more stable, fully differentiable, and often take less than 100 training samples to achieve better results.

For most realistic scenarios where more than 100 training samples are available, MLP models are the most promising. They are among the top predictors on HW-NAS-Bench and demonstrate outstanding performance on the TransNAS-Bench-101 datasets. We found that specialized architecture encodings are primarily beneficial for little training data but suspect that they enjoy an additional advantage when network topologies are more complex and diverse (White et al., 2021).

While the query time for all predictors is less than $0.05$s and thus negligible, there is a notable difference in training time (see Appendix F), primarily due to the hyper-parameter optimization. We recommend Ridge Regression for very little amounts of training data and LGBoost otherwise if that is an important factor.

If a NAS method selects architectures based on hardware metric predictions, we strongly suggest verifying the results by measuring the true metric value afterward. Doing so may eliminate inferior candidates and improve the average result substantially. Finally, if the limiting factor to a NAS method is the slow measurement of hardware metrics, using a much faster predictor may lead to an improvement, even if the prediction model is less accurate.

## 7 CONCLUSIONS

This work evaluated various hardware-metric prediction models on ten problems of different metrics, devices, and network architecture types. We then simulated the selection process for different test set sizes and predictor inaccuracies to improve our understanding of predictor-based architecture selection. We find that even imperfect predictors may improve NAS if their low query time enables considering more candidate architectures. Finally, verifying the predictions for the selected candidates can lead to a drastic improvement of their average performance. The code and results are made available, thus acting both for recommendation and as a baseline for future works.

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

## A  BEST PRACTICES FOR NAS, CODE AND DATA

To improve the reproducibility and facilitate fair experimental comparisons, we follow the best-practices checklist (Lindauer & Hutter, 2020):

- **Release Code for the Training Pipeline(s) you use.** Our experiments are based on White et al. (2021), who use NASLib (Ruchte et al., 2020) to compare 31 methods for accuracy prediction. Our NASLib fork, extending the framework for HW-NAS-Bench, TransNAS-Bench, some performance predictors and the hypervolume simulations, is provided in the supplementary materials. We intend to either make our fork available on GitHub or submit the changes upstream once this paper is accepted/published.

- **Use the Same Evaluation Protocol for the Methods Being Compared.** Aside from the implementation of each predictor, all experiments use the same pipeline.

- **Validate The Results Several Times.** We ran each predictor 50 times, with seeds $\{0, ..., 49\}$. The reductions in hypervolume are simulated 1000 times using different a different subset of the data set, for each combination of {iteration, HW-NAS data set, noise on HW metric}.

- **Control Confounding Factors.** While all experiments used the same software libraries and hardware resources, they were run on different machines to speed up the evaluation. We found hardly any benefit in using a GPU even for the network-based predictors, which is why every method only used two CPU cores. The OS is Ubuntu 18.04, notable software packages are PyTorch 1.9.0, numpy 1.19.5, scikit-learn 0.24.2, pybnn 0.0.5, ngboost 0.3.11, and xgboost 1.4.2

- **Report the Use of Hyperparameter Optimization.** See Appendix C.

In addition to the code in the supplementary materials, we also provide the experimental results as csv files. Running the predictors and hypervolume simulations takes some time, but the easy access to the data of the finished experiments may prove useful for future research. Please see *readme.md* in the accompanying code zip file for instructions.

## B  ENCODINGS AND PREDICTORS

### B.1  DATA ENCODINGS

Every architecture $a \in \mathcal{A}$ requires a unique representation, which depends on the used predictor. The common encoding types are:

**Adjacency one-hot**: Each architecture $a$ is uniquely defined by the chosen candidate operation on every path. For example, each architecture in NAS-BENCH-201 consists of a repeated cell structure, which has five candidate operations on each of the six paths. Therefore there are $5^6 = 15625$ unique architectures, which can each be referenced by a sequence of operation-indices such as $[0\ 1\ 2\ 3\ 4\ 0]$. Many predictors perform better if the sequence is presented as a one-hot encoding, which is in this case $[10000\ 01000\ 00100\ 00010\ 00001\ 10000]$.

Similarly, the **path-encoding** (used by BANANAS) is a one-hot representation over the used candidate operation all possible paths. Since the connectivity within cells for HW-NAS-Bench and TransNAS-Bench-101 is fixed, it provides no more information than the adjacency one-hot encoding. If the connectivity can be adjusted more freely, as in the NAS-Bench-101 search space, the additional information may improve the fit.

The encodings for **BONAS**, **GCN**, and **NAO** each provide further information in addition to the Adjacency one-hot vector, most notably the adjacency-matrix. This $\{0, 1\}^{(N+2)\times(N+2)}$ matrix lists describes which of the $N$ architecture paths (rows) serves as inputs for each other path (column), and also includes input/output.

## B.2 PREDICTORS

We briefly describe the 18 predictor methods in our experiments. We adopt their implementations from the NASLib library (see Appendix A), which we extend with Linear Regression, Ridge Regression, and Support Vector Machines from the scikit-learn package; and a simple Lookup Table implementation. Unless specified otherwise, the methods use the adjacency one-hot encoding.

- **BANANAS** An ensemble of three MLP models with five to 20 layers, each using the path-encoding (White et al., 2019).

- **Bayesian Linear Regression** A bayesian model that assumes (1) a linear dependency between inputs and outputs, and (2) that the samples are normally distributed (Bishop, 2007).

- **BOHAMIANN** A bayesian inference predictor using stochastic gradient Hamiltonian Monte Carlo (SGHMC) to sample from a bayesian neural network (Springenberg et al., 2016).

- **BONAS** Bayesian Optimization for NAS (Shi et al., 2020) uses a GCN predictor within an outer loop of bayesian optimization, as a meta-learning task. The GCN requires encoding the adjacency matrix of each architecture.

- **Gaussian Process** A simple model that assumes a joint Gaussian distribution underlying the training data (Rasmussen, 2003).

- **GCN** A Graph Convolutional Network that makes use of an adjacency-matrix encoding of each architecture (Wen et al., 2020).

- **Linear Regression** A simple model that assumes an independent value/cost for each operation/layer, which only need to be summed up. Unlike the Lookup Table model, it uses a least-square fit on the training data.

- **Lookup Table** The most simple and perhaps widely used model for differentiable architecture selection. It generally assumes a single baseline architecture (e.g. [001 001] in adjacency one-hot encoding), and a lookup matrix $\mathbb{R}^{(\text{num layers}) \times (\text{num candidates})}$ that contains the increases/reductions in the metric for each layer and candidate operation. The metric value of a new architecture can be predicted with a simple sum over the respective matrix entries and the baseline value. The model is obtained from measuring either each candidate operation in isolation, or by computing the differences between the baseline architecture and specific variations (e.g. [010 001] or [100 001], to measure the first candidates). This model always requires $1 + (\text{num layers}) \cdot (\text{num candidates} - 1)$ neighbored architectures to fit. We detail the resulting correlation values for each used dataset in Appendix E.

- **LGBoost** Light Gradient Boosting Machine (LightGBM or LGBoost, Ke et al. (2017)) is a lightweight gradient-boosted decision tree model.

- **MLP** We use fully-connected Multi Layer Perceptrons in two size-categories.

- **NAO** NAO (Luo et al., 2018) uses an encoder-decoder topology, which encodes/compresses an architecture to a continuous representation, and decodes it again. This representation is further used to make architecture predictions.

- **NGBoost** Natural Gradient Boosting (NGBoost, Duan et al. (2020)) is a gradient-boosted decision tree model that uses natural gradients to estimate uncertainty.

- **Ridge Regression** Ridge Regression (Saunders et al., 1998) extends the Linear Regression least-squares fit with a regularization term that serves as bias-variance tradeoff.

- **Random Forests** An ensemble of decision trees (Liaw et al., 2002).

- **Sparse Gaussian Process** an approximation of Gaussian Processes that summarizes training data (Candela & Rasmussen, 2005).

- **Support Vector Machine** A model that maps its inputs to a high-dimensional space, where training samples are used as support-vectors for decision-boundaries (Cortes & Vapnik, 1995).

- **XGBoost** eXtreme Gradient Boosting (XGBoost, Chen & Guestrin (2016)) is a gradient-boosted decision tree model.

## C HYPERPARAMETERS

We list our default and hyper-parameter sample ranges in Table 2. For comparability with White et al. (2021), we only change the values of newly introduced parameterized predictors: Ridge Regression, Support Vector Machines, and small MLPs.

| Model | Hyper-parameter | Range/Choice | Log-transform | Default |
|---|---|---|---|---|
| BANANAS | Num. Layers | [5, 25] | false | 20 |
| | Layer width | [5, 25] | false | 20 |
| | Learning rate | [0.0001, 0.1] | true | 0.001 |
| BONAS | Num. Layers | [16, 128] | true | 64 |
| | Batch size | [32, 256] | true | 128 |
| | Learning rate | [0.00001, 0.1] | true | 0.0001 |
| GCN | Num. Layers | [64, 200] | true | 144 |
| | Batch size | [5, 32] | true | 7 |
| | Learning rate | [0.00001, 0.1] | true | 0.0001 |
| | Weight decay | [0.00001, 0.1] | true | 0.0003 |
| LGBoost | Num. leaves | [10, 100] | false | 31 |
| | Learning rate | [0.001, 0.1] | true | 0.05 |
| | Feature fraction | [0.1, 1] | false | 0.9 |
| MLP (small) | Num. layers | [2, 5] | false | 3 |
| | Layer width | [16, 128] | true | 32 |
| | Learning rate | [0.0001, 0.1] | true | 0.001 |
| | Activation function | {relu, tanh, hardswish} | | relu |
| MLP (huge) | Num. layers | [5, 25] | false | 20 |
| | Layer width | [5, 25] | false | 20 |
| | Learning rate | [0.0001, 0.1] | true | 0.001 |
| NAO | Num. layers | [16, 128] | true | 64 |
| | Batch size | [32, 256] | true | 100 |
| | Learning rate | [0.00001, 0.1] | true | 0.001 |
| NGBoost | Num. estimators | [128, 512] | true | 64 |
| | Learning rate | [0.001, 0.1] | true | 0.081 |
| | Max depth | [1, 25] | false | 6 |
| | Max features | [0.1, 1] | false | 0.79 |
| Ridge Regression | Regularization $\alpha$ | [0.25, 2.5] | false | 1.0 |
| Random Forests | Num. estimators | [16, 128] | true | 116 |
| | Max features | [0.1, 0.9] | true | 0.17 |
| | Min samples (leaf) | [1, 20] | false | 2 |
| | Min samples (split) | [2, 20] | true | 2 |
| Support Vector Machine | Regularization $C$ | [0.5, 1.5] | false | 1.0 |
| | Kernel | {linear, poly, rbf, sigmoid} | | rbf |
| XGBoost | Max depth | [1, 15] | false | 6 |
| | Min child weight | [1, 10] | false | 1 |
| | Col sample (tree) | [0, 1] | false | 1 |
| | Learning rate | [0.001, 0.5] | true | 0.3 |
| | Col sample (level) | [0, 1] | false | 1 |

Table 2: Hyper-parameter ranges and default values of the configurable predictors

# D  NAS-BENCH-201 / HW-NAS-BENCH CELL DESIGN

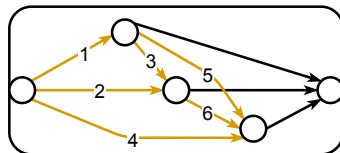

Figure 5: Basic NAS-Bench-201 / HW-NAS cell design. Each of the six orange paths is finalized with exactly one out of five candidate operations {Zero, Skip, Convolution 1×1, Convolution 3×3, Average Pooling 3×3}.

# E  SELECTION OF DATASETS

| | Linear Regression | | | | | | | | | | XGBoost | LUT |
|---|---|---|---|---|---|---|---|---|---|---|---|---|
| | 11 | 25 | 55 | 124 | 276 | 614 | 1366 | 3036 | 6748 | 15000 | 15000 | - |
| ImageNet16-120-raspi4_latency | 0.324 | 0.205 | 0.606 | 0.676 | 0.705 | 0.716 | 0.715 | 0.723 | 0.728 | 0.729 | 0.757 | 0.443 |
| cifar100-pixel3_latency | 0.392 | 0.292 | 0.732 | 0.780 | 0.797 | 0.803 | 0.806 | 0.809 | 0.812 | 0.812 | 0.877 | 0.484 |
| cifar10-edgegpu_latency | 0.370 | 0.258 | 0.724 | 0.790 | 0.806 | 0.819 | 0.820 | 0.822 | 0.830 | 0.829 | 0.926 | 0.175 |
| cifar100-edgegpu_energy | 0.376 | 0.275 | 0.732 | 0.793 | 0.812 | 0.821 | 0.821 | 0.823 | 0.831 | 0.831 | 0.920 | 0.221 |
| ImageNet16-120-eyeriss arith. int. | 0.369 | 0.293 | 0.748 | 0.805 | 0.817 | 0.827 | 0.825 | 0.832 | 0.843 | 0.846 | 0.970 | 0.861 |
| cifar10-pixel3_latency | 0.388 | 0.300 | 0.733 | 0.780 | 0.797 | 0.805 | 0.805 | 0.810 | 0.813 | 0.813 | 0.878 | 0.475 |
| cifar10-raspi4_latency | 0.393 | 0.315 | 0.740 | 0.787 | 0.799 | 0.805 | 0.807 | 0.810 | 0.813 | 0.813 | 0.890 | 0.462 |
| cifar100-raspi4_latency | 0.393 | 0.308 | 0.744 | 0.786 | 0.801 | 0.807 | 0.810 | 0.810 | 0.814 | 0.814 | 0.888 | 0.445 |
| ImageNet16-120-pixel3_latency | 0.398 | 0.312 | 0.739 | 0.786 | 0.799 | 0.807 | 0.809 | 0.812 | 0.815 | 0.816 | 0.884 | 0.509 |
| cifar100-edgegpu_latency | 0.375 | 0.268 | 0.728 | 0.793 | 0.810 | 0.821 | 0.820 | 0.822 | 0.831 | 0.831 | 0.924 | 0.191 |
| cifar10-edgegpu_energy | 0.375 | 0.284 | 0.728 | 0.792 | 0.810 | 0.821 | 0.823 | 0.824 | 0.831 | 0.831 | 0.922 | 0.183 |
| ImageNet16-120-edgegpu_energy | 0.377 | 0.281 | 0.733 | 0.797 | 0.814 | 0.825 | 0.825 | 0.826 | 0.834 | 0.833 | 0.926 | 0.280 |
| ImageNet16-120-edgegpu_latency | 0.379 | 0.264 | 0.737 | 0.799 | 0.817 | 0.826 | 0.826 | 0.828 | 0.836 | 0.835 | 0.938 | 0.277 |
| cifar10-eyeriss arith. int. | 0.384 | 0.296 | 0.757 | 0.811 | 0.826 | 0.835 | 0.832 | 0.843 | 0.854 | 0.854 | 0.969 | 0.826 |
| cifar100-eyeriss arith. int. | 0.384 | 0.297 | 0.757 | 0.811 | 0.826 | 0.835 | 0.833 | 0.844 | 0.855 | 0.856 | 0.971 | 0.830 |
| ImageNet16-120-fpga_latency | 0.443 | 0.494 | 0.904 | 0.936 | 0.947 | 0.951 | 0.948 | 0.951 | 0.952 | 0.952 | 0.983 | 0.965 |
| ImageNet16-120-fpga_energy | 0.443 | 0.494 | 0.905 | 0.935 | 0.947 | 0.951 | 0.948 | 0.951 | 0.952 | 0.952 | 0.983 | 0.965 |
| ImageNet16-120-eyeriss_latency | 0.457 | 0.937 | 0.953 | 0.954 | 0.954 | 0.954 | 0.953 | 0.953 | 0.954 | 0.954 | 0.952 | 0.989 |
| cifar10-eyeriss_latency | 0.461 | 0.943 | 0.959 | 0.959 | 0.960 | 0.960 | 0.959 | 0.960 | 0.960 | 0.960 | 0.958 | 0.995 |
| cifar100-eyeriss_latency | 0.462 | 0.946 | 0.963 | 0.963 | 0.963 | 0.963 | 0.963 | 0.963 | 0.964 | 0.963 | 0.962 | 0.998 |
| cifar10-eyeriss_energy | 0.456 | 0.967 | 0.985 | 0.985 | 0.985 | 0.985 | 0.985 | 0.985 | 0.985 | 0.985 | 0.975 | 0.996 |
| ImageNet16-120-eyeriss_energy | 0.458 | 0.967 | 0.985 | 0.985 | 0.986 | 0.985 | 0.986 | 0.985 | 0.985 | 0.986 | 0.972 | 0.998 |
| cifar100-eyeriss_energy | 0.457 | 0.967 | 0.985 | 0.985 | 0.985 | 0.986 | 0.985 | 0.986 | 0.986 | 0.986 | 0.976 | 0.998 |
| cifar10-fpga_energy | 0.458 | 0.973 | 0.987 | 0.987 | 0.987 | 0.987 | 0.987 | 0.987 | 0.987 | 0.987 | 0.986 | 0.999 |
| cifar100-fpga_energy | 0.458 | 0.973 | 0.987 | 0.987 | 0.987 | 0.987 | 0.987 | 0.987 | 0.987 | 0.987 | 0.986 | 0.999 |
| cifar100-fpga_latency | 0.457 | 0.973 | 0.987 | 0.987 | 0.987 | 0.987 | 0.987 | 0.987 | 0.987 | 0.987 | 0.986 | 0.999 |
| cifar10-fpga_latency | 0.457 | 0.973 | 0.987 | 0.987 | 0.987 | 0.987 | 0.987 | 0.987 | 0.987 | 0.987 | 0.986 | 0.999 |

Table 3: Kendall's Tau test correlation for Linear Regression, XGBoost, and Lookup Table (LUT) on all HW-NAS-Bench datasets (rows), for different amounts of available training data (columns), tested on the remaining 625 samples. The Lookup Table model is tested on all 15625 architectures. We selected the five data sets at the top.

| | Linear Regression | | | | | | | | | | XGBoost | LUT |
|---|---|---|---|---|---|---|---|---|---|---|---|---|
| | 9 | 18 | 34 | 65 | 123 | 234 | 442 | 837 | 1585 | 2999 | 2999 | - |
| jigsaw | 0.201 | 0.227 | 0.410 | 0.535 | 0.586 | 0.605 | 0.616 | 0.624 | 0.631 | 0.632 | 0.661 | 0.201 |
| class_object | 0.268 | 0.262 | 0.518 | 0.646 | 0.711 | 0.741 | 0.759 | 0.771 | 0.780 | 0.780 | 0.828 | 0.701 |
| room_layout | 0.275 | 0.271 | 0.527 | 0.653 | 0.721 | 0.753 | 0.768 | 0.780 | 0.789 | 0.789 | 0.896 | 0.685 |
| class_scene | 0.275 | 0.268 | 0.527 | 0.653 | 0.721 | 0.755 | 0.768 | 0.782 | 0.789 | 0.790 | 0.907 | 0.710 |
| segmentsemantic | 0.282 | 0.259 | 0.545 | 0.684 | 0.746 | 0.780 | 0.798 | 0.809 | 0.816 | 0.818 | 0.871 | 0.726 |

Table 4: Kendall's Tau test correlation for Linear Regression and XGBoost on the five used TransNAS datasets (rows), for different amounts of available training data (columns), tested on the remaining 256 samples. The Lookup Table model (LUT) is tested on all 3256 architectures.

**HW-NAS-Bench:**  To select five datasets that are (1) non-linear and (2) different from one another, we first fit Linear Regression to every available dataset, with the results listed in Table 3. The bottom 12 datasets can be accurately fit with only 25 training samples, so they are not very interesting as a

challenge. On these datasets, the Lookup Table model achieves exceptional performance. Since the networks for CIFAR10, CIFAR100 and ImageNet16-120 only differ slightly, their measurements on the same device and metric (e.g. raspi4 latency) is very similar. To improve the generalizability of our results, we thus select datasets on different devices and metrics, which are listed at the top of Table 3. As displayed in Figure 6, their data distributions are generally different.

**TransNAS-Bench-101:** Since the latency measurements of the architectures is generally very similarly distributed (see Figure 7), it is not necessary to train the predictors on all of them. We select all data sets that provide the *test_loss* and *inference_time* attributes for all architectures, resulting in exactly the five datasets listed in Section 4 (the other two datasets contain more specific test losses).

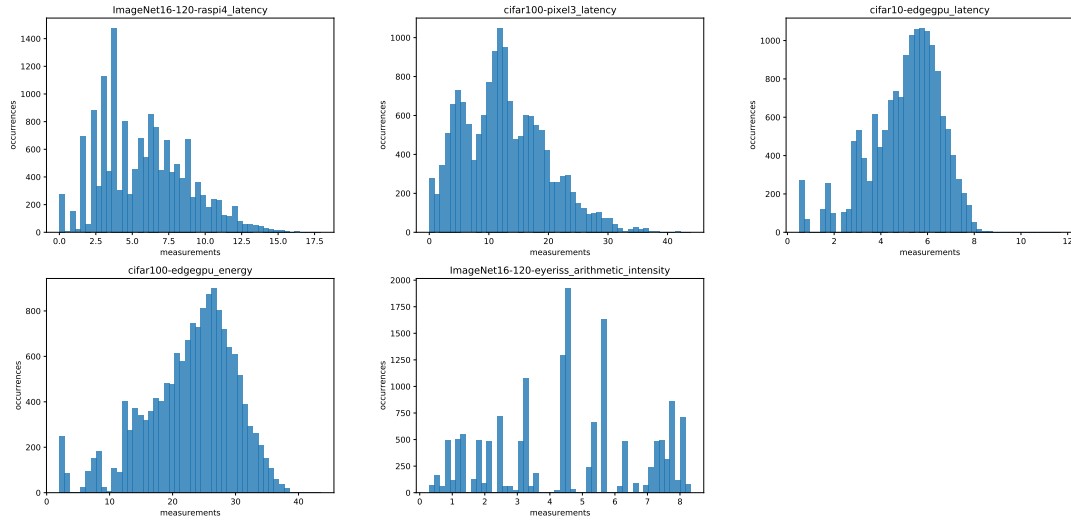

Figure 6: How the data of each selected HW-NAS-Bench dataset is distributed (not normalized).

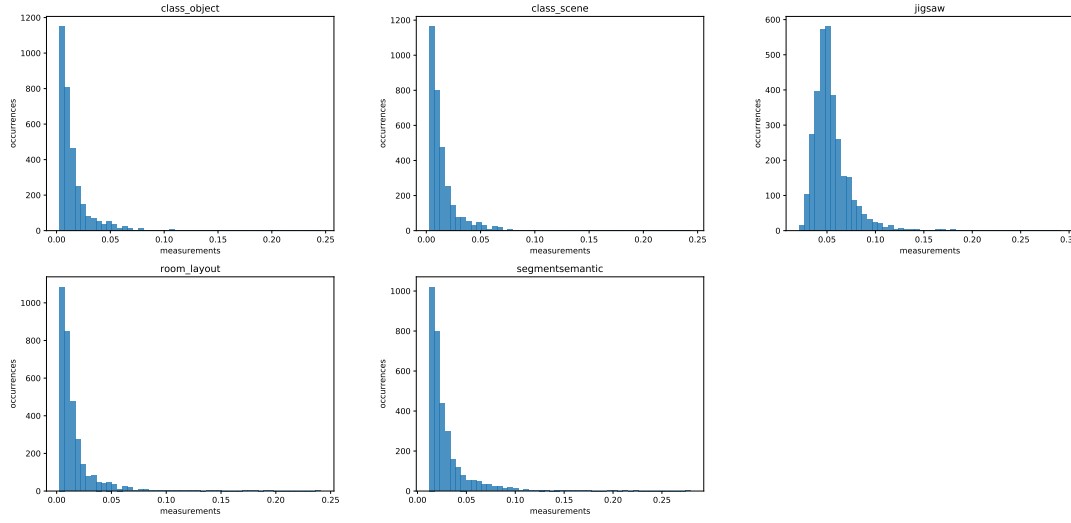

Figure 7: How the data of each selected TransNAS-Bench-101 dataset is distributed (not normalized). Since all architectures are measured for latency on the same hardware, the resulting datasets are much less diverse than the HW-NAS-Bench ones.

## F  PREDICTOR FIT TIME

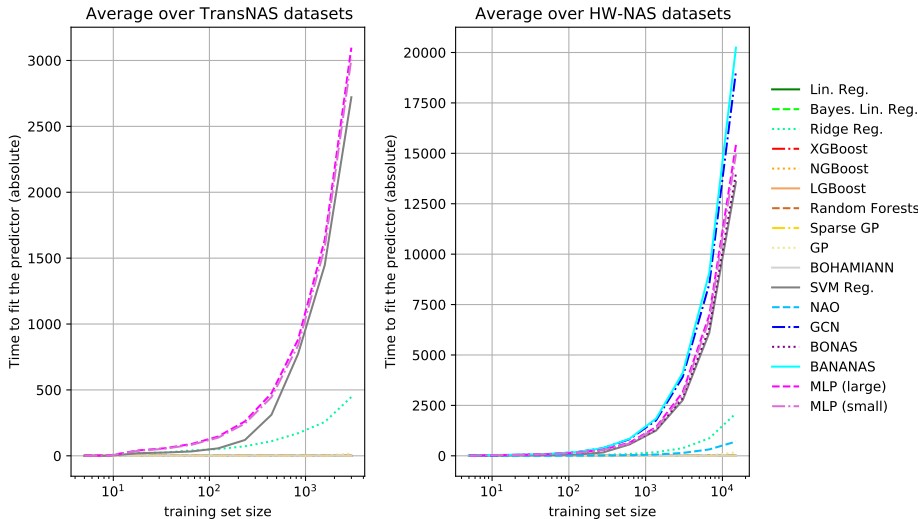

Figure 8: Fit time (in seconds) of predictors to data, depending on the training set size. By far the most expensive methods are network-based. However, a significant portion of this time is spent on the hyper-parameter optimization prior to the actual fitting.

## G  APPROXIMATING PREDICTOR MISTAKES

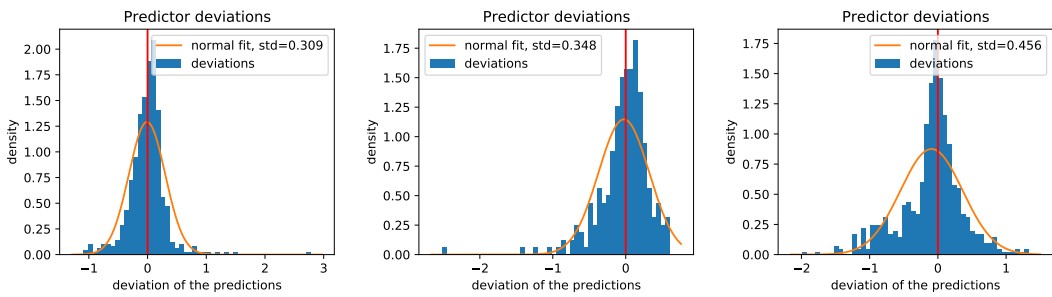

Figure 9: Further examples of predictor deviation distributions, as visualized in the center of Figure 2. **Left:** Linear Regression on CIFAR100, edgegpu, energy consumption. **Center:** Support Vector Machine on Jigsaw. **Right:** small MLP on ImageNet16-120, raspi4, latency.

Intuitively, the predictor deviation distributions (see Figures 2 and 9) generally resemble a normal distribution. However, most predictors:

(1) Have a notable peak, sometimes off-center (e.g. at x=0.2)

(2) Have less density than a normal distribution almost everywhere else

(3) Have some outliers (e.g. at x>1.5) that are extremely unlikely for a normal distribution

We measured the p-value for different distributions on the first 100 test samples using a T-Test, every time we evaluated a predictor. The average statistics can be found in Table 5. Since a large number of empirical observations generally pushes the p-value to 0, this only serves to compare them to each other. We find that the outliers (3) appear often enough and are so unlikely to happen for a normal distribution, that even a uniform distribution has a higher statistical support. Consequentially, we approximate the common predictor deviations by sampling from a mixed distribution that adresses (1) to (3).

|         | p-value |
|---------|---------|
| normal  | 0.028   |
| cauchy  | 0.030   |
| lognorm | 0.028   |
| t       | 0.028   |
| uniform | 0.037   |

Table 5: P-values of different distributions, trying to fit the distribution of all predictor mistakes according to a t-test. Larger values are better, but comparing many empirically sampled points with a true density function tends to push the p-values to 0.

This mixed distribution consists of two Normal distributions ($N_1$, $N_2$) and one Uniform distribution ($U$), from which we sample with 72.5%, 26.5% and 1% respectively. For some constant $v$:

- We uniformly sample a shift $c$ from $[0, 2 \cdot v]$, that is used to push the centers of $N_1$ and $N_2$ to $x > 0$ and $x < 0$ respectively.
- We sample each value from $N_1(c, v)$, $N_2(-c, 3 \cdot v)$, and $U_1(-15 \cdot v, 15 \cdot v)$ randomly, with the weighting given above.
- We normalize (subtract mean, divide by standard deviation) our sampled distribution and then scale it to the desired standard deviation.
- The predictors produce non-smooth distributions. We simulate that by sampling 15 times fewer values as needed, and repeat them as often.

The code for the simulation is also provided (see Appendix A). As seen in Figure 10, the resulting simulated deviation distributions generally resemble a common predictor pattern. We do not account for differences in predictors, training set sizes or more, since that may become too specific and over-engineered.

Appendix I visualizes simulation sanity checks. We find that the simulation is slightly pessimistic and simplified, but resembles the results of actual predictors.

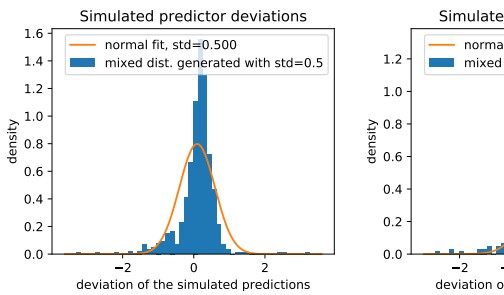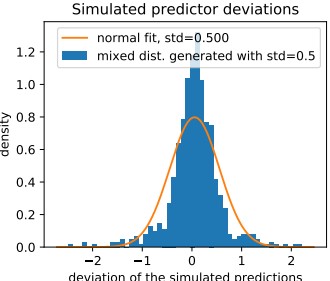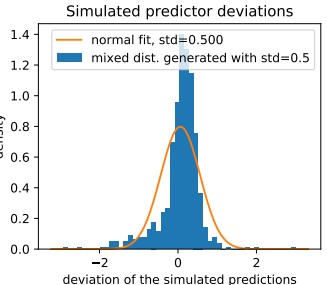

Figure 10: The sampled values of gaussian+uniform fit the measured predictor mistakes better than a single distribution, as they are roughly normally distributed, but include outliers.

# H  MEASURING SIMULATED MISTAKES

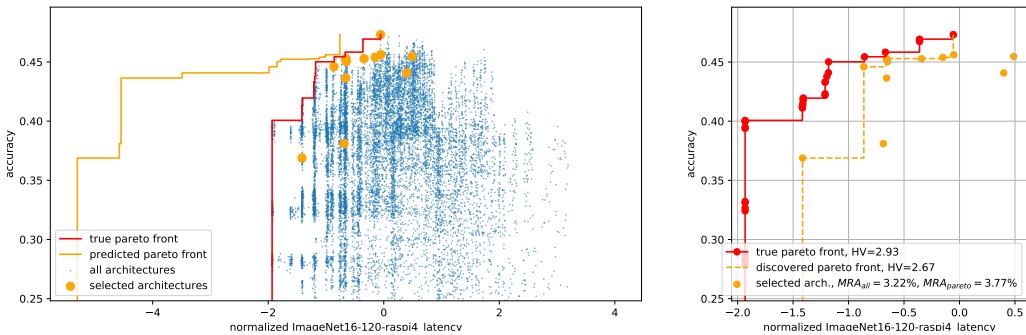

Figure 11: Similar to Figure 3. When the discovered Pareto set is considerably worse than the true Pareto set, it is possible for the Mean Reduction of Accuracy of the Pareto subset ($MRA_{pareto}$) to be *worse* than the average over all architectures ($MRA_{all}$). This naturally happens more frequently for worse predictors with a high sampling std. and low KT correlation. Consequentially, the difference between $MRA_{all}$ and $MRA_{pareto}$ is wider for better predictors (see Figure 4). Additionally, all of the selected non-Pareto-front members are clustered in a high-latency area and redundant with each other. This emphasizes the limitations of just considering drops in accuracy, as the hardware metric aspect is ignored. In this case, the predictor-guided selection failed to find a low-latency solution. In this regard, hypervolume is a better but less intuitive metric.

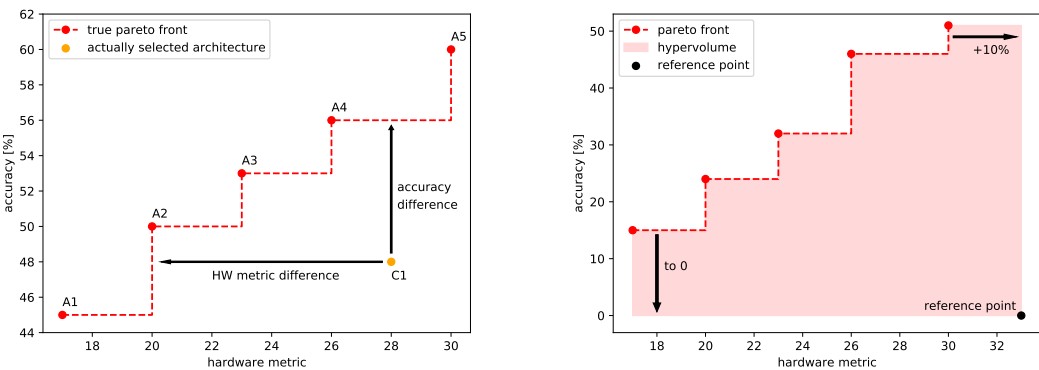

Figure 12: Examples to explain measurement methods.
**Left:** The distance of each selected candidate architecture C1 to the true Pareto front is measured, for accuracy and the hardware metric. C1 is dominated by A2, A3, and A4 of the true Pareto set. A2 has a slightly higher accuracy than C1 while being much better on the hardware metric, e.g. latency. A4 has a slightly better hardware metric value, but much higher accuracy. Given several candidate architectures, their differences are averaged.
**Right:** We compute the reference point for the hypervolume (for two objectives: area under a Pareto front) by multiplying the highest hardware metric value from the true Pareto front with 1.1, and accuracy 0. While we are consistent throughout all experiments, this choice is arbitrary, as there is no obviously correct choice for the reference point. If the hypervolume of a supposed Pareto front is computed, the reference point of the true Pareto front is reused. Thus, choosing inferior architectures will always reduce the hypervolume. We arbitrarily chose the multiplier of $m = 1.1$ as a middle ground between making the rightmost point of the Pareto front irrelevant ($m = 1.0$) and overemphasizing it ($m >> 1.0$).

## I   SIMULATION SANITY CHECK

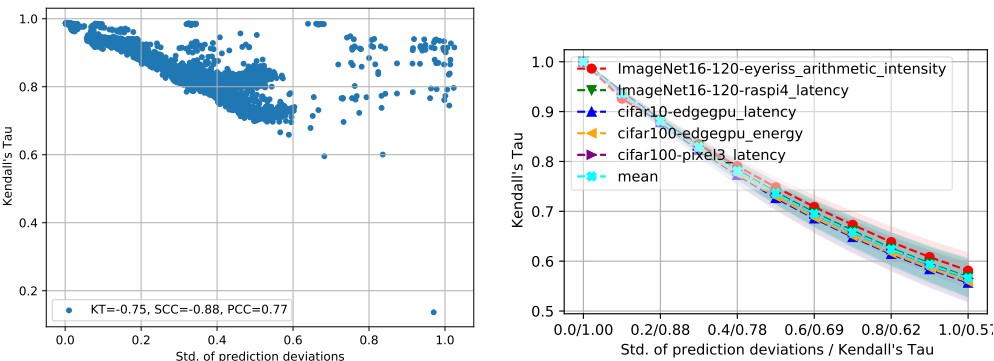

Figure 13: Standard deviation over the predictor deviations (x axis) and Kendall's Tau correlation (y axis), for the trained predictors on HW-NAS-Bench (left) and in simulation (right). The simulated predictor inaccuracies are slightly pessimistic (low KT), but still match the true values.

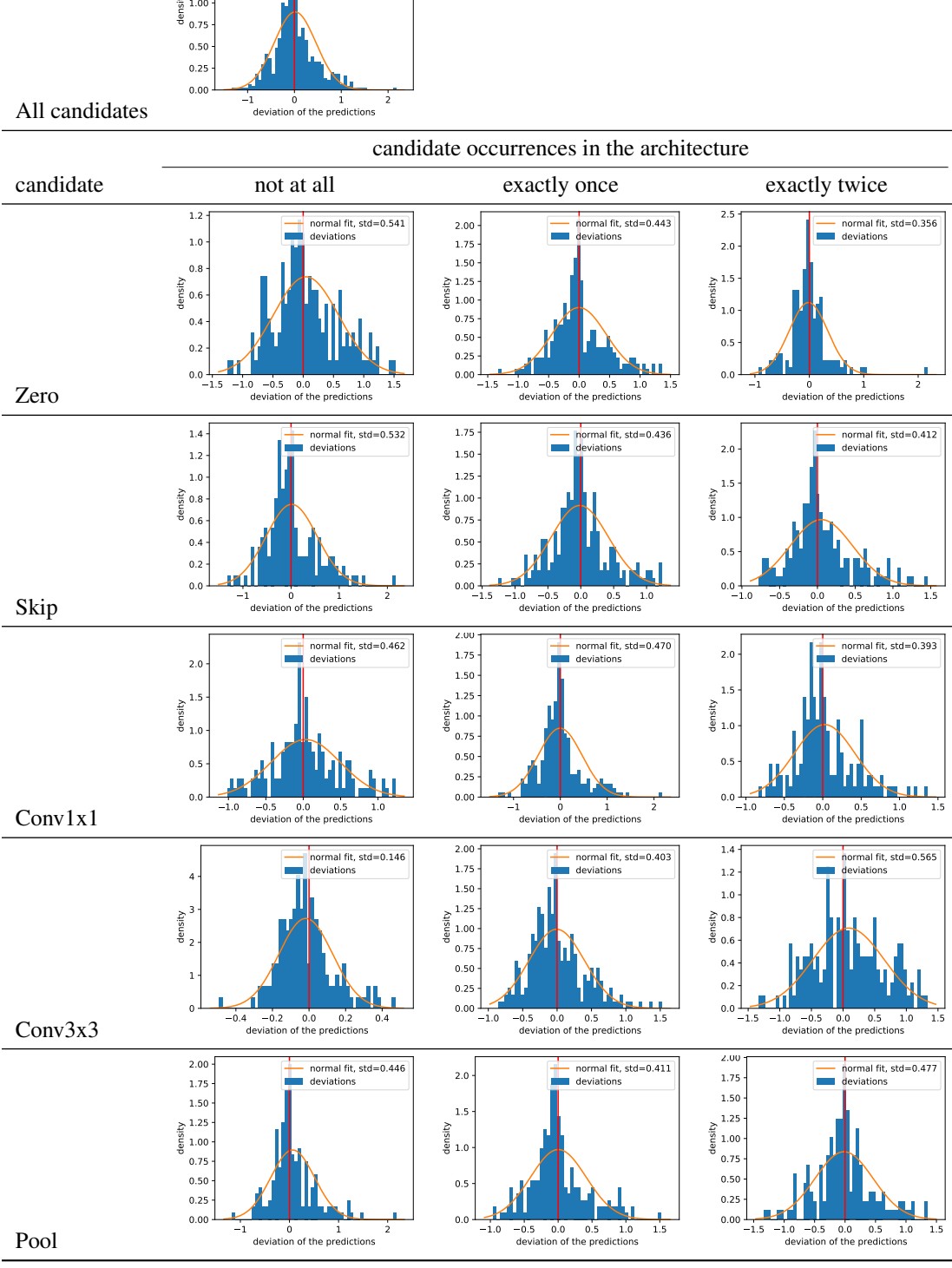

Table 6: How a trained XGB predictor deviates from the ground-truth values for different architecture subsets, akin to Figure 2. While they are not exactly the same, they still resemble the distribution over the entire test set (top plot, 625 samples). One noteworthy exception is when no Conv3x3 operations are used at all, in which case the standard deviation is considerably smaller.

