# OpenReview forum: "What to expect of hardware metric predictors in NAS"
_ICLR.cc/2022/Conference — ICLR 2022 Submitted_

### Official Review · Reviewer_GUHu · 2021-11-02

**Correctness:** 3
**Technical Novelty And Significance:** 1
**Empirical Novelty And Significance:** 4
**Recommendation:** 6
**Confidence:** 4

**Details Of Ethics Concerns:**

No ethical concerns.

**Main Review:**

Strengths
- This is an important problem. Although there have been experimental studies on regular performance predictors, not much was known about hardware metric predictors until this paper, which is a more realistic setting.
- The analysis is good. For example, Fig 1 (right) and Fig 2 (right) give easily visualizable data by centering on predictor-average. Also, they focused on analyzing only the most challenging prediction tasks, which seems beneficial.
- The authors have thorough appendices where they release all of their code, raw results, and methodologies. So reproducibility and ease of follow-up is high.
- It is interesting that in Section 6, the authors simulate the predictor (now there is a prediction of a prediction) to speed up results. However, I have some follow-up questions about this in “weaknesses”.

Weaknesses
- The authors state a few interesting insights at the end of Section 4, 5, 6. For example, MLP works great on HW-NAS-Bench and TransNAS-Bench-101, and the widely used lookup table does not perform nearly as well. Although experimental survey papers are very useful, the best ones follow up on something that is surprising, interesting, or insight-provoking. For example, the authors could go a step further and look into why MLP performed so well, or, find some other type of insight in their study that leads to new directions.

I also have a few smaller nitpicks and questions about the experiments:
 - Section 6 is nice, but don’t a lot of NAS methods use a predictor-guided search such as BO, where the predictor is constantly being updated (even for hardware aware NAS algorithms)? So it would also be interesting to see how the predictors compare within NAS.
 - Section 6 is hard to follow. I don’t fully understand how the predictors are simulated even when reading the figures and Appendix G. It seems like the authors are focusing on approximating the distribution of mistakes. But then do you draw from the distribution on each prediction? What if the mistakes are one-sided based on properties of the architecture? Maybe I am misunderstanding.
 - In Section 6, it sounds like the authors estimated the hardware metrics but “All accuracy values remain accurate.” In the context of real NAS, isn’t this unrealistic because hardware metrics are much easier to get than the accuracies? Why not also use a predictor for the accuracy, and then compute the error of the Pareto-front after that? Or maybe I am misunderstanding?
 - When the authors explained in Section 4 that they only selected five out of 27 datasets, I thought this was totally arbitrary, especially when looking that they are imbalanced (e.g., CIFAR 10 only used once, latency used three times while energy and intensity are used once). But then eventually I got to Appendix E and realized that it is a principled selection, and some of the 27 are trivial and not worth analyzing. So I suggest that the authors stress this methodology more in Section 4. Also, it could still be useful to add some analysis on all 27, in addition to keeping the experiments that are already there. Or, some aggregate analysis about which metrics are hardest and easiest, which edge devices, which image datasets (e.g., like a summary of Table 2).
- Also on page 6, there are a few statements that the authors should have citations for: "Latency is generally the most crucial hardware metric" and "the macro-level search space is more applicable to most real-world problems than a micro-level search space."

**Summary Of The Paper:**

In neural architecture search (NAS), performance predictors are an important tool, because predicting attributes such as accuracy can save costly measurements. Hardware metric predictors predict metrics such as latency in addition to accuracy. This paper gives an empirical study of 18 different performance predictors on NAS-Bench-201 and TransNAS-Bench-101, evaluating the rank correlation of the predictors on different datasets and predicted metrics. Then they evaluate how the inaccuracy of the predictors affect the predicted Pareto-front of architectures, compared to the true Pareto front. The authors run this experiment by simulating the errors in predictors. The authors find that MLP models perform the best on average across all settings they tried, and leads to an acceptable predicted Pareto-front.

**Summary Of The Review:**

Overall, this paper is important and useful for the community, and has pretty good and thorough analysis. However, it could have more interesting insights, and there are some questions about Section 6. Overall, I currently will give a weak reject and look forward to the authors’ response.

---

> ### Author Response · Authors · 2021-11-15
> **reply part 1/3**
>
> Thank You for Your valuable time and helpful feedback.
>
> We have added an overview comment detailing the changes of our new paper version,
> our detailed replies to your review are found below:
>
>
> > ####Main Review:
> >
> >
> > Strengths
> >
> > - This is an important problem. Although there have been experimental studies
> > on regular performance predictors, not much was known about hardware metric
> > predictors until this paper, which is a more realistic setting.
> >
> > - The analysis is good. For example, Fig 1 (right) and Fig 2 (right)
> > give easily visualizable data by centering on predictor-average.
> > Also, they focused on analyzing only the most challenging prediction tasks,
> > which seems beneficial.
> >
> > - The authors have thorough appendices where they release all of their code,
> > raw results, and methodologies. So reproducibility and ease of follow-up is high.
>
> Thank You for Your kind words.
> We noticed that, although hardware predictors are widely used in NAS,
> a study with this context in mind was missing.
> Even more so, while ranking correlation metrics allow us to compare predictors,
> what specific values actually meant for the selection of architectures was an
> open but fundamental question.
> To provide clear and reproducible answers (which are often missing, see e.g. [5]), we decided to be as open as possible
> with our methods, code, and results.
>
> We hope that our study leads other researchers to replace Lookup Table models with
> other predictors, which might already improve many results.
> In fact, on the evaluated HW-NAS-Bench data sets, choosing other prediction models would improve the
> mean accuracy of the selected architectures by over 2% (see the new Figure 4, left).
>
>
>
> > - It is interesting that in Section 6, the authors simulate the predictor
> > (now there is a prediction of a prediction) to speed up results.
> > However, I have some follow-up questions about this in “weaknesses”.
>
> Speeding up to consider a larger sample size is only one of the reasons.
> The more important one is that is simply not possible to obtain some predictors with specific
> qualities from fitting alone. An obvious example the evaluation of a nearly perfect predictor
> (std=0.1, KT=~0.94) on a test set with size >=15000. Training a predictor to this quality is
> not possible with only a few remaining samples
> (at most 625, since HW-NAS-Bench has only 15625 architectures).
> As the simulation is not constrained by such difficulties, a more comprehensive
> study becomes possible.
>
>
>
>
>
> > Weaknesses
> >
> > - The authors state a few interesting insights at the end of
> > Section 4, 5, 6. For example, MLP works great on HW-NAS-Bench and
> > TransNAS-Bench-101, and the widely used lookup table does not perform
> > nearly as well. Although experimental survey papers are very useful,
> > the best ones follow up on something that is surprising, interesting,
> > or insight-provoking. For example, the authors could go a step further and
> > look into why MLP performed so well, or, find some other type of insight
> > in their study that leads to new directions.
>
> We agree that many predictor results are not surprising, except perhaps for the poor performance
> of Lookup Tables.
> The most important insights come from combining these results with those from the simulation.
> We know that an MLP with a Kendall's Tau correlation of ~0.9 is better than Linear Regression
> with ~0.81, but can not anticipate what that means for a NAS algorithm.
> According to the simulation results, the difference between these two prediction models causes
> optimal NAS methods to select architectures that differ by almost 0.7% accuracy.
> We have rewritten major parts of Section 6 (now Section 5) and hope that the text now
> clearly reflects this context.

---

> > ### Author Response · Authors · 2021-11-15
> > **reply part 2/3**
> >
> >
> > > I also have a few smaller nitpicks and questions about the experiments:
> > >
> > > - Section 6 is nice, but don’t a lot of NAS methods use a predictor-guided
> > > search such as BO, where the predictor is constantly being updated
> > > (even for hardware aware NAS algorithms)?
> > > So it would also be interesting to see how the predictors compare within NAS.
> >
> > On-line methods (such as BO, evolutionary algorithms, reinforcement learning, ...) are indeed
> > common in NAS. However, they do not operate on the same problem. BO is used to find some
> > architecture of, e.g., maximum accuracy and minimum latency. It will learn on-line to solve
> > this problem, by querying triplets of (architecture, accuracy, latency).
> > On the other hand, the hardware metric predictors provide the required latency values.
> >
> > While we could compare BO/EAs/RL and more based on the simulation, this was not the intended goal.
> > The simulation always uses the optimal architectures (given the imperfect hardware predictions)
> > and is therefore an upper bound that such methods may approach.
> > Conclusions drawn from the simulation are statements how NAS behaves if the
> > optimal architectures are discovered, given different levels of predictor-quality
> > and number of evaluated architectures.
> > We hope that these points are clearer in the rephrased sections.
> >
> >
> >
> > > - Section 6 is hard to follow. I don’t fully understand how the predictors
> > > are simulated even when reading the figures and Appendix G.
> > > It seems like the authors are focusing on approximating the distribution
> > > of mistakes. But then do you draw from the distribution on each prediction?
> > > What if the mistakes are one-sided based on properties of the architecture?
> > > Maybe I am misunderstanding.
> >
> > You are correct, we simulate predictors based on how they deviate from the ground-truth values.
> > In practice, to create the simulated-predicted values we simply add randomly drawn values
> > from the distribution (Appendix G) to the ground-truth values. Naturally, the simulated-predicted
> > values are created anew for every single simulation.
> > We hope that the rewritten paper text is also clearer on these points now.
> >
> > Whether specific architecture properties affect the predictor deviations is a fair question.
> > We added various plots to the
> > simulation sanity check in Appendix I to visualize the distributions for different architecture subsets.
> > While we find that they differ mostly in how closely most values are distributed around the center,
> > such extremes do not occur.
> > We agree that such considerations could make the simulation more correct, but do not think that
> > the likely minor improvement justifies the added complexity.
> >
> >
> >
> > > - In Section 6, it sounds like the authors estimated the hardware metrics
> > > but “All accuracy values remain accurate.” In the context of real NAS,
> > > isn’t this unrealistic because hardware metrics are much easier to
> > > get than the accuracies? Why not also use a predictor for the accuracy,
> > > and then compute the error of the Pareto-front after that?
> > > Or maybe I am misunderstanding?
> >
> > You are absolutely correct that hardware metrics are easier to obtain than accuracy values
> > and that NAS methods often use predictors for both. They are usally independent
> > (e.g. super-network for accuracy, Lookup Table for the hardware metric),
> > which allows us to evaluate them in isolation.
> > While we initially considered simulating the architecture selections for different degrees
> > of accuracy-correctness, we later decided not to introduce the additional complexity.
> > We have stressed these points more in the respective paper text now, since it was not clear
> > before.
> >
> >
> >
> >
> > > - When the authors explained in Section 4 that they only selected five out of
> > > 27 datasets, I thought this was totally arbitrary, especially when
> > > looking that they are imbalanced (e.g., CIFAR 10 only used once, latency
> > > used three times while energy and intensity are used once).
> > > But then eventually I got to Appendix E and realized that it is a
> > > principled selection, and some of the 27 are trivial and not worth analyzing.
> > > So I suggest that the authors stress this methodology more in Section 4.
> >
> > We agree that the initial phrasing was not clear enough on the problem and our
> > data set selection. We have rewritten the paragraph in the newly uploaded version and hope that it is
> > clearer now.

---

> > > ### Author Response · Authors · 2021-11-15
> > > **reply part 3/3**
> > >
> > >
> > > > Also, it could still be useful to add some analysis on all 27,
> > > > in addition to keeping the experiments that are already there.
> > > > Or, some aggregate analysis about which metrics are hardest and easiest,
> > > > which edge devices, which image datasets (e.g., like a summary of Table 2).
> > >
> > > Thank You for this suggestion. We have added an aggregate device/metric table to the
> > > predictor results Section, which shows that the difficulty of fitting predictors depends primarily
> > > on the hardware device. We also find that Lookup Tables perform excellent on FPGA and Eyeriss
> > > metrics, but only marginally better than Linear Regression with only 124 training samples.
> > > In all other cases, they are considerably worse.
> > >
> > >
> > >
> > > > - Also on page 6, there are a few statements that the authors should have
> > > > citations for: "Latency is generally the most crucial hardware metric"
> > > > and "the macro-level search space is more applicable to most real-world
> > > > problems than a micro-level search space."
> > >
> > > We admit that this statement was unintentionally biased towards image classification,
> > > where many modern NAS methods optimize a MobileNetV2-like topology for accuracy and latency (e.g. [1] or [2]).
> > > Such networks are also used as backbones for object detection (e.g. [3]), where latency
> > > is especially crucial for problems such as self-driving cars and pedestrian detection.
> > > We agree that this bias does not necessarily hold true for different problem types such
> > > as the construction of recurrent cells (e.g. [4]).
> > > We have rephrased this sentence to be more clear.
> > >
> > > > ####Summary Of The Review:
> > > >
> > > > Overall, this paper is important and useful for the community, and has pretty good and thorough analysis. However, it could have more interesting insights, and there are some questions about Section 6. Overall, I currently will give a weak reject and look forward to the authors’ response.
> > >
> > > Thank You for Your kind words. We have rewritten major parts of Section 6 (now Section 5)
> > > to clarify the issues and further detail the most important points.
> > >
> > >
> > >
> > >
> > > Thank You for Your time and helpful feedback,
> > >
> > > First Author
> > >
> > >
> > >
> > > [1] EfficientNet: Rethinking Model Scaling for Convolutional Neural Networks, https://arxiv.org/abs/1905.11946v3
> > >
> > > [2] HardCoRe-NAS: Hard Constrained diffeRentiable Neural Architecture Search, https://arxiv.org/abs/2102.11646
> > >
> > > [3] EfficientDet: Scalable and Efficient Object Detection, https://arxiv.org/abs/1911.09070
> > >
> > > [4] Efficient Neural Architecture Search via Parameter Sharing, https://arxiv.org/abs/1802.03268
> > >
> > > [5] NAS evaluation is frustratingly hard, https://arxiv.org/abs/1912.12522

---

> > > > ### Comment · Reviewer_GUHu · 2021-11-18
> > > > **Thank you for your response.**
> > > >
> > > > Thanks for the thorough replies to my review and the other reviews, and thanks for the major update to the paper.
> > > >
> > > > I particularly like the new plots in Appendix I to sanity check the simulation, the new aggregate device/metric tables, and the updates to Sections 4 and 5 which have improved the writing in the paper.
> > > >
> > > > I am increasing my score from 5 to 6.
> > > >
> > > > I have one main follow-up comment. The authors defended their decision not to compare hardware metric predictors within NAS (e.g. inside BO / evolutionary algorithms), saying that using the ground-truth accuracy gives an upper bound, and that conclusions drawn from the simulation are statements how NAS behaves if the optimal architectures are discovered.
> > > >
> > > > I still think that it would be most useful to practitioners if experiments were added that run BO / evolution / etc with different hardware predictors, and compare the quality of the Pareto front.

---

### Official Review · Reviewer_bHy6 · 2021-11-03

**Correctness:** 4
**Technical Novelty And Significance:** 2
**Empirical Novelty And Significance:** 3
**Recommendation:** 6
**Confidence:** 5

**Main Review:**

**Strength:**
1. The paper is well-written with clear logical flow and easy to follow.
2. The evaluation across different predictors and tasks is solid and the codes are provided.
3. The simple but effective insights can benefit the NAS community.

**Weakness:**
1. The major concern of this paper is the technical novelty since no new technique is proposed and the provided insights are relatively intuitive. In particular, a less accurate but fast predictor can intuitively lead to better results if it can see more architectures, where the measurement cost is assumed to be the bottleneck. More novel insights are expected.
2. The averaged correlation in Fig. 1 across different devices (and tasks) may not be the only metric for evaluating the predictors, i.e., the device-specific correlation is also important. For example, as demonstrated in HW-NAS-Bench, for some search spaces (i.e., a sequential space like FBNet space) and many devices, the operation-wise hardware performances are additive, indicating a lookup table will be accurate enough as a predictor. For most deployment scenarios with determined types of devices, such device-specific discussions could strength the contributions of this paper and provide more detailed insights for the community.
3. The discussions about the influences of the predictors to the architecture selection should be NAS-algorithm-specific while such discussions are missing in the paper. For example, for reinforcement learning based NAS methods, the hardware measurement is independent of the subnetwork training, while the hardware performance will participate in the training process of FBNet as a regularization term, where an inaccurate hardware performance predictor will more directly affect the results after convergence. Such NAS-algorithm-specific discussions may also guide the NAS algorithm selection given affordable hardware performance predictors in addition to predictor selections.





**Summary Of The Paper:**

This paper systematically evaluates a wide range of hardware performance predictors across different networks/devices/tasks and analyzes the influence of such predictors to the architecture selection process. It provides insights for the community about how to select a proper hardware performance predictor in different scenarios.

**Summary Of The Review:**

Given the limited novelty and lacked discussions elaborated in the weakness part, I tend to deem this paper marginally below the acceptance threshold. I'm willing to adjust my scores if the concerns are properly addressed.

---

> ### Author Response · Authors · 2021-11-15
> **reply**
>
> Thank You for Your valuable time and helpful feedback.
>
> We have added an overview comment detailing the changes of our new paper version,
> our detailed replies to your review are found below:
>
> > ####Strength:
> >
> > - The paper is well-written with clear logical flow and easy to follow.
> > - The evaluation across different predictors and tasks is solid and the codes
> > are provided.
> > - The simple but effective insights can benefit the NAS community.
>
>
>
> > ####Weakness:
> >
> > - The major concern of this paper is the technical novelty since no new
> > technique is proposed and the provided insights are relatively intuitive.
> > In particular, a less accurate but fast predictor can intuitively lead
> > to better results if it can see more architectures,
> > where the measurement cost is assumed to be the bottleneck.
> > More novel insights are expected.
>
> While we think that a large-scale predictor study is useful to build better NAS models,
> we agree that the novelty is questionable.
> Nonetheless, the detailed study of how hardware predictions affect the NAS results is
> (to the best of our knowledge) the first of its kind.
> Since the important relationship between the predictor results and the simulation
> was not clear enough, we have rewritten major parts of these Sections.
> We hope that the new text and the added examples now properly emphasize the important insights
> from the simulation.
>
>
>
> > - The averaged correlation in Fig. 1 across different devices (and tasks)
> > may not be the only metric for evaluating the predictors,
> > i.e., the device-specific correlation is also important.
> > For example, as demonstrated in HW-NAS-Bench, for some search spaces
> > (i.e., a sequential space like FBNet space) and many devices,
> > the operation-wise hardware performances are additive, indicating a
> > lookup table will be accurate enough as a predictor.
> > For most deployment scenarios with determined types of devices,
> > such device-specific discussions could strength the contributions of
> > this paper and provide more detailed insights for the community.
>
> We agree that device-specific statistics are very helpful for some use cases.
> While Appendix E and the provided results csv files already contain the relevant information,
> we also see that the information is not easily accessible that way.
> Based on Your suggestion and those of another reviewer,
> we have added an aggregate metric/device table to our predictor results in Section 4.
>
> The table clearly shows that the difficulty of fitting predictors depends on the device,
> not with the metric, and that Lookup Tables are decent on the FPGA and Eyeriss devices.
> On the other hand, we can not confirm that they are a good model for the sequential search
> space of TransNAS-Bench-101 on Tesla V100 GPUs.
> HW-NAS-Bench itself provides only a Lookup Table model for the FBNet space,
> not the results of measured architectures, so that cross-reviewing
> this finding immediately is not possible.
>
>
> > - The discussions about the influences of the predictors to the architecture
> > selection should be NAS-algorithm-specific while such discussions are
> > missing in the paper. For example, for reinforcement learning based NAS methods,
> > the hardware measurement is independent of the subnetwork training,
> > while the hardware performance will participate in the training process
> > of FBNet as a regularization term, where an inaccurate hardware
> > performance predictor will more directly affect the results after convergence.
> > Such NAS-algorithm-specific discussions may also guide the NAS algorithm
> > selection given affordable hardware performance predictors in addition to
> > predictor selections.
>
> Our current simulation assumes that the used multi-objective NAS method selects a set of
> pareto-optimal architectures, as done by evolutionary algorithms and other approaches.
> Since the simulation will always select the best candidates (given the imperfect predictions),
> the results are actually an upper bound on how well NAS can perform.
> We absolutely agree that studying how such predictors affect NAS methods based on gradients,
> reinforcement learning, and more is an important topic in future research.
>
>
>
>
> > ####Summary Of The Review:
> >
> > Given the limited novelty and lacked discussions elaborated in the weakness part,
> > I tend to deem this paper marginally below the acceptance threshold.
> > I'm willing to adjust my scores if the concerns are properly addressed.
>
> We hope that the rewritten paper Sections and comments here properly address your points.
> If that is not the case, please let us know.
>
>
>
>
> Thank You for Your time and helpful feedback,
>
> First Author

---

### Official Review · Reviewer_bBS4 · 2021-11-07

**Correctness:** 4
**Technical Novelty And Significance:** 1
**Empirical Novelty And Significance:** 2
**Recommendation:** 5
**Confidence:** 2

**Main Review:**

Strength:
- This work did a very comprehensive survey across different kinds of the performance predictors and across two existing hardware performance dataset. The network architectures considered are also well rounded.
- This work gives reasonable explanations/insights for most of the phenomena in the conducted experiments.

Weakness:
- Novelty is limited
- The insights are shallow and obvious
- Unclear on the aspects of how to counter the imperfect predictions on guiding the network architecture selection



**Summary Of The Paper:**

This work provides a compressive analysis across multiple (18) different hardware performance predictors by
- (1) collecting their performance under different amounts of training data and different input network structures and showing each prediction method’s advantageous/disadvantageous scenarios
- (2) analyzing the  prediction accuracy's influence on selecting subsequent hardware architecture and giving the insights on how to pick/design hardware performance predictors for the NAS.

Based on the observation drawn, MLP ones are found to be the most promising predictors in terms of the accuracy with limited training samples, while the Lookup Table model serves as a very cheap and straightforward guidance. In terms of the architecture selection guided by the model-based predictor, the work conjectures by verifying some of the selected network  structures’  hardware performance and increase the number of explored network structures can lead to better pareto front of the selected network architecture

**Summary Of The Review:**

- **Novelty is limited**: Most of this work is simply using the existing hardware performance predictors and running them to collect results on the existing hardware dataset. Too much focus on introducing the setup and configurations of these hardware performance predictors and dataset, and simply numerating the results from the conducted experiment, while the proposed insights are overshadowed.

- **The insights are shallow and obvious**: Even though the paper presents multiple insights from the experiments, most of them are rather shallow and obvious. For instance, the paper stressed that the model-based can benefit more from more training data as compared to the lookup table based predictors, which is rather obvious as model based is data driven while lookup table is deterministic. However, some specific necessary and useful  insights are missing. For instance for page 5, most of the texts are devoted to reporting which methods perform better and which methods scale better  with more data. The underlying reasons or hypotheses are not provided.

- **Unclear on the aspects of how to counter the imperfect predictions on guiding the network architecture selection**: One of the most interesting aspects about this work is that it proposes imperfect predictions from the hardware predictors can also lead to decent selected architectures. However, it is rather unclear how to achieve this and why. For instance, in page 7, the authors mention “verifying the hardware metric predictions of the 13 architectures”; which 13 architectures? It was never mentioned elsewhere. I really hope the author can stress this part more.

---

> ### Author Response · Authors · 2021-11-15
> **reply part 1/2**
>
> Thank You for Your valuable time and helpful feedback.
>
> We have added an overview comment detailing the changes of our new paper version,
> our detailed replies to your review are found below:
>
> > Based on the observation drawn, MLP ones are found to be the most promising
> > predictors in terms of the accuracy with limited training samples,
> > while the Lookup Table model serves as a very cheap and straightforward guidance.
> > In terms of the architecture selection guided by the model-based predictor,
> > the work conjectures by verifying some of the selected network structures’
> > hardware performance and increase the number of explored network structures
> > can lead to better pareto front of the selected network architecture
>
> We want to emphasize that, according to our results, the widely used Lookup Tables
> can only compete when very few training samples are available for any data-driven predictor.
>
>
> > ####Main Review:
> >
> >
> > Strength:
> >
> > - This work did a very comprehensive survey across different kinds of the
> performance predictors and across two existing hardware performance dataset.
> The network architectures considered are also well rounded.
> >
> > - This work gives reasonable explanations/insights for most of the phenomena
> in the conducted experiments.
>
>
> >
> > Summary Of The Review:
> >
> > - Novelty is limited: Most of this work is simply using the existing hardware
> > performance predictors and running them to collect results on the
> > existing hardware dataset. Too much focus on introducing the setup and
> > configurations of these hardware performance predictors and dataset,
> > and simply numerating the results from the conducted experiment,
> > while the proposed insights are overshadowed.
>
> We have combined Sections 4 and 5 to considerably reduce the required text for these experiments.
> This allowed us to add a small aggregate table for results across metrics and device and go into
> slightly more detail why different predictors behave like that.
> We were also able to elaborate on the simulation in more detail and added examples that connect
> both topics.
> We hope that the important insights and novelty of this paper are now properly demonstrated.
>
>
>
> > - The insights are shallow and obvious: Even though the paper presents
> > multiple insights from the experiments, most of them are rather shallow and obvious.
> > For instance, the paper stressed that the model-based can benefit more
> > from more training data as compared to the lookup table based predictors,
> > which is rather obvious as model based is data driven while lookup table
> > is deterministic. However, some specific necessary and useful insights are missing.
> > For instance for page 5, most of the texts are devoted to reporting which
> > methods perform better and which methods scale better with more data.
> > The underlying reasons or hypotheses are not provided.
>
> We agree that many predictor results are as expected. We have compressed this text and added
> remarks why the training set size affects the models differently.
> However, the novelty of the paper lies primarily in Section 5 (previously Section 6).
> The simulation allows us to see what a Kendall's Tau correlation of 0.8 actually means
> for NAS algorithms, and what happens when the number of considered architectures is changed.
> This is the first study to answer or quantify such questions.
>
> Since we failed to properly convey the connections and insights, we have rewritten the related
> text according to the feedback and included multiple examples.
>
>
>
> > - Unclear on the aspects of how to counter the imperfect predictions on
> > guiding the network architecture selection:
> > One of the most interesting aspects about this work is that it proposes
> > imperfect predictions from the hardware predictors can also lead to decent
> > selected architectures. However, it is rather unclear how to achieve this and why. [...]
>
> As an example, consider using ground-truth latency values for a random subset of 1000 architectures,
> and imperfect latency predictions for 2000 architectures.
> In the right of Figure 4 (previously Figure 5), we see that a predictor with a Kendall's Tau
> ranking correlation of only ~0.65 is enough for a NAS method to select architectures of equivalent quality.
>
> Intuitively, if more architectures are considered, the density close to their true pareto front is higher.
> Architectures further away from the pareto front are only selected if a predictor wrongly
> assigns them decent predictions (e.g. for latency), which are unmatched by any actually-better
> candidate.
> However, increasing the density reduces the chances for that to happen.

---

> > ### Author Response · Authors · 2021-11-15
> > **reply part 2/2**
> >
> >
> >
> > > [...] For instance, in page 7, the authors mention “verifying the hardware metric
> > > predictions of the 13 architectures”; which 13 architectures? It was never
> > > mentioned elsewhere. I really hope the author can stress this part more.
> >
> > The 13 selected architectures can be seen in Figure 3, they are the pareto-optimal architectures in
> > the considered set (given the imperfect hardware predictions). By actually measuring their
> > hardware metrics (verifying the results), not just trusting the predictions,
> > we learn that some of them are suboptimal.
> >
> > The rewritten Section 5 (previously Section 6) should describe the simulation approach more clearly and
> > properly connect it to the predictor results now.
> > Please continue to ask if we failed to properly clarify Your questions, or if the new text does
> > not answer them.
> >
> >
> >
> > Thank You for Your time and helpful feedback,
> >
> > First Author

---

### Official Review · Reviewer_7VPE · 2021-11-07

**Correctness:** 4
**Technical Novelty And Significance:** 1
**Empirical Novelty And Significance:** 2
**Recommendation:** 6
**Confidence:** 4

**Main Review:**

While I think that the motivation to have a large-scale study of hardware metric predictors similar as White et al. do for performance predictors is a reasonable one, however I think that this paper needs to improve its presentation and clarity. There is a lot of text (e.g. in Sections 3, 4 and 5) which might be compressed or moved to the appendix, adding space for additional experiments to be conducted. Considering that the novelty and the results presented in the paper are not surprising, I think a more thorough empirical evaluation is necessary.

These are some of my questions and recommendations:

- I am quite confused by the experiments and results in Section 6. In Figure 4, how do you obtain the simulated pareto front? Do you evaluate all the architectures in the space using the predictor (for both accuracy and hardware metric)?
- What are the "selected architectures"? Are they the architectures in the "simulated pareto front" evaluated on the true HW-NAS-Bench?
- The main purpose of tabular NAS benchmarks, including HW-NAS-Bench that is for hardware metrics, is to provide a fast way to prototype and evaluate NAS algorithms, which will ultimately be evaluated on real world benchmarks. Therefore, I think, for the case at hand, it would be useful to evaluate the trained predictors on a setting where they are used in a multi-objective NAS algorithm together with the performance predictors.
- I found many of the plots to be redundant, e.g. the right-hand side plots in Fig 1 and 2. I think the authors can gain some space by moving them to the appendix and adding additional experimental results in the main paper.

**Summary Of The Paper:**

This paper provides a large-scale study of hardware metric predictors on 2 recent tabular benchmarks: Trans-NAS-Bench-101 and HW-NAS-Bench. The authors compare the predictors in terms of rank correlation and their ability to simulate the true pareto front and demonstrate that model-based predictors outperform Lookup Tables with very few training data points and that simple MLPs can outperform the more specialized models such as GNNs.

**Summary Of The Review:**

I think that this paper is useful since it evaluates model-based predictors on the same open-source framework, enabling a fair comparison of different predictors. However, I think that the experimental evaluation conducted by the authors is not enough in order for me to give an acceptance score. There is also quite some work to be done in order to improve the structuring and clarity of the text in my opinion.

---

> ### Author Response · Authors · 2021-11-15
> **reply part 1/2**
>
> Thank You for Your valuable time and helpful feedback.
>
> We have added an overview comment detailing the changes of our new paper version,
> our detailed replies to your review are found below:
>
> > ####Main Review:
> >
> > While I think that the motivation to have a large-scale study of hardware metric
> > predictors similar as White et al. do for performance predictors is a
> > reasonable one, however I think that this paper needs to improve its
> > presentation and clarity. There is a lot of text
> > (e.g. in Sections 3, 4 and 5) which might be compressed or moved
> > to the appendix, adding space for additional experiments to be conducted.
>
> Based on Your feedback, and that of the other reviewers, we have made significant changes
> in our updated version. Please see the overview post for details.
>
>
> > Considering that the novelty and the results presented in the paper
> > are not surprising, I think a more thorough empirical evaluation is necessary.
>
> We agree that many predictor results are as expected. The poor performance of Lookup Tables
> is interesting nonetheless, especially considering their widespread use.
> We have added an aggregate table of metrics/devices based on other feedback, which may also
> be interesting in this regard.
>
> We also added multiple examples in Section 5 (previously Section 6) that properly connect
> the simulation to the predictors and emphasize the important insights.
> Please let us know if You feel that we have not properly addressed this point.
>
>
>
>
> >
> > ####These are some of my questions and recommendations:
> >
> > - I am quite confused by the experiments and results in Section 6.
> > In Figure 4, how do you obtain the simulated pareto front?
> > Do you evaluate all the architectures in the space using the predictor
> > (for both accuracy and hardware metric)?
> >
> > - What are the "selected architectures"?
> > Are they the architectures in the "simulated pareto front" evaluated on the true
> > HW-NAS-Bench?
>
> The predictors make a prediction for every architecture in the test set.
> While their ground-truth values are plotted in blue (Figure 3), which is matched by the ground-truth
> pareto front in red, the imperfect predictors suggest a different setting.
> According to the predicted values, the 13 architectures marked with orange dots provide
> the optimal trade-off between accuracy and the hardware metric (here: latency).
> Based on their predictions, the pareto front is much more shifted to the lower latency values,
> as visualized by the orange line (''predicted pareto front'').
>
> A NAS algorithm would select these 13 architectures, since they are believed to be best.
> As seen in the right of Figure 3, while they are close, they are not optimal.
> This is quantified by a reduction in accuracy and hypervolume.
> The results in Figure 4 build on such simulations, evaluating the cost of imperfect
> predictions in various configurations.

---

> > ### Author Response · Authors · 2021-11-15
> > **reply part 2/2**
> >
> >
> > > - The main purpose of tabular NAS benchmarks,
> > > including HW-NAS-Bench that is for hardware metrics,
> > > is to provide a fast way to prototype and evaluate NAS algorithms,
> > > which will ultimately be evaluated on real world benchmarks. [...]
> >
> > While we fully agree that evaluating on real-world problems is desirable, the costs are
> > infeasible.
> > The reason is that evaluating a selected architecture requires training it to completion,
> > which may take 10 days on e.g. ImageNet. Preferably, multiple training sessions are averaged
> > for more stable results.
> > If each multi-objective NAS search selects just 10 architectures, and we also require many searches to
> > average out the accuracy predictors, hardware predictors, and NAS method randomness, we quickly
> > approach many thousand GPU hours.
> > Additionally, it is also not possible to compare the NAS-discovered architectures with the
> > unknown pareto set.
> >
> > Therefore, we evaluated the predictors in a simulation based on HW-NAS-Bench (Section 5, previously Section 6).
> > While not as interesting as a real-world problem, at least a detailed evaluation is possible.
> >
> >
> > > [...] Therefore, I think, for the case at hand,
> > > it would be useful to evaluate the trained predictors on a setting where they
> > > are used in a multi-objective NAS algorithm together with the performance predictors.
> >
> > We admit that our simulation results do not consider different multi-objective NAS algorithms
> > or inaccurate accuracy values. The reasons are:
> > - In simulation, the search process always selects the pareto optimal architectures,
> > given the inaccurate hardware metric predictions. It is an upper bound how well any particular
> > multi-objective method can perform.
> > Comparing whether different methods approach this upper bound is an interesting question,
> > but was not the intention of our simulation.
> > - The simulation uses the ground-truth accuracy values, since they are obtained from an
> > independent model (usually a super-network for accuracy, e.g. lookup tables for latency).
> > Since they are independent, they can also be optimized in isolation.
> > We initially considered also varying the correctness of the accuracy values, but decided that the
> > gain does not justify the increased complexity.
> >
> > We admit that our initial paper version did not explain these points.
> > We hope that the rewritten Section 5 is much clearer.
> >
> >
> >
> > > - I found many of the plots to be redundant,
> > > e.g. the right-hand side plots in Fig 1 and 2.
> > > I think the authors can gain some space by moving them to the appendix and
> > > adding additional experimental results in the main paper.
> >
> > While we agree that they do not contain additional information, we think that the alternate
> > representation (in Figure 1 and 2) are both beneficial. The absolute
> > predictor performance on the left is the main result and useful in combination with Figure 5,
> > the relative performance on the right makes a visual comparison much easier.
> >
> > >
> > > ####Summary Of The Review:
> > >
> > > I think that this paper is useful since it evaluates model-based predictors
> > > on the same open-source framework, enabling a fair comparison of different predictors.
> > > However, I think that the experimental evaluation conducted by the authors
> > > is not enough in order for me to give an acceptance score. [...]
> >
> > We agree that adding further datasets from additional benchmarks is desirable for a broader
> > comparison. We only used HW-NAS-Bench and TransNAS-Bench-101 for the following reasons:
> >
> > Firstly, they already cover a variety of architecture designs (purely sequential and multi-path)
> > and candidate operations, which even cause the TransNAS-Bench-101 networks to have
> > different layer-wise tensor sizes.
> >
> > Secondly, aside from HW-NAS-Bench, detailed hardware- and architecture-specific are scarce.
> > Most benchmarks only report such metrics (mostly latency) on a single device, if at all.
> > The reason is that the main intention of most NAS benchmarks is to correctly estimate the
> > loss or accuracy, not specific hardware metrics.
> >
> > Finally, our  most important and novel contribution are the insights from the simulation.
> > We see that were not clear how the predictor results and the simulation are connected.
> > The new text contains some examples that emphasize their relationship, and show how different
> > prediction models have a real impact on the NAS results.
> >
> >
> > > [...] There is also quite some work to be done in order to improve the
> > > structuring and clarity of the text in my opinion.
> >
> > We have uploaded a new paper version with significant changes to the structure and text,
> > which are detailed in an overview post.
> > If your concerns are not addressed, please inform us so that we can clear up further details.
> >
> >
> >
> > Thank You for Your time and helpful feedback,
> >
> > First Author

---

> > > ### Comment · Reviewer_7VPE · 2021-11-22
> > > **Thank you for incorporating the suggestions**
> > >
> > > I thank the authors for addressing some of my concerns. I am increasing my score to 6, however I still think that it would be interesting to evaluate the hardware predictors inside various NAS algorithms. Reviewer GUHu made the same suggestion.

---

### Author Response · Authors · 2021-11-15
**general comments and changelog**

Dear reviewers,

thank You for Your valuable time, questions, and suggestions.
Based on Your feedback we have made several changes to the paper. Most importantly, we
compressed the predictor results,
added connecting examples between the predictor results and the novel simulation insights, and hopefully
made it overall clearer to read.

While we agree that evaluating and comparing different prediction models is not itself novel,
the gained information directly benefits existing and future NAS methods.
Furthermore, the results are also important in combination with our novel simulation, which quantifies
what to expect from hardware metric predictions in context of NAS.
The test set size, predictor (in)correctness, and data set have measurable effects,
which may be of interest for the design of future algorithms.

More detailed changelog:

- Sections 4 and 5 (predictor results on HW-NAS-Bench and TransNAS-Bench-101) are now combined
in the new Section 4. This allowed us to compress the text for their results.
- We stessed our HW-NAS-Bench dataset selection more clearly (Section 4).
- We hinted at underlying reasons why different models benefit differently from an increase
in training data (Section 4).
- We added an aggregate device/metric table that, showing that the difficulty of fitting predictors
is a property of the hardware device. The data was already available in Appendix E, but
not as accessible (Section 4).
- We stressed the advantages of a simulation more clearly (Section 5), and how it works.
- We added some examples that connect the simulation results to the predictor results (Section 5).
- We fixed the cropping of few images in the Appendix, which cut off text before.
- We fixed a bug in our code, which caused the Lookup Table models to disregard the last
candidate operation (this only affected few architectures). Interestingly, the bug made the
model perform better.


With kind regards,

First Author

---

### Decision · Program_Chairs · 2022-01-20

**Decision:**

Reject

**Comment:**

This paper comprehensively evaluated 18 different performance predictors on ten combinations of metrics, devices, network types, and training tasks for NAS. While evaluating and comparing different prediction models is not itself novel, the authors provided many insights that are potentially interesting to future NAS developments.

Reviewer reactions to this paper are rather mediocre and lukewarm. It is in general consensus that this work gives a good empirical analysis on hardware metric predictors for NAS, but the novelty is low and it is perhaps a bit incremental (e.g., nothing "shockingly new" was revealed, and observations are mostly "as expected"). Despite the authors improving the paper during rebuttal with new plots/tables, there remain to be unaddressed comments, e.g., adding experiments that run BO / evolution / etc with different hardware predictors and comparing the quality of the Pareto front. Those missed points were also raised in the private discussion.

After personally reading this paper, AC sides with most reviewers that this paper lacks true novelty nor technical excitement. While the empirical study is valuable, it perhaps suits venues other than ICLR, e.g., the NeurIPS benchmark track.